# Single-cell RNA-seq reveals fibroblast heterogeneity and increased mesenchymal fibroblasts in human fibrotic skin diseases

Cheng-Cheng Deng [1,5], Yong-Fei Hu [1,2,5], Ding-Heng Zhu[1,5], Qing Cheng[1], Jing-Jing Gu[1], Qing-Lan Feng[1], Li-Xue Zhang[1], Ying-Ping Xu[1], Dong Wang [1,2], Zhili Rong [1,3,4 ✉] & Bin Yang [1 ✉]

Fibrotic skin disease represents a major global healthcare burden, characterized by fibroblast hyperproliferation and excessive accumulation of extracellular matrix. Fibroblasts are found to be heterogeneous in multiple fibrotic diseases, but fibroblast heterogeneity in fibrotic skin diseases is not well characterized. In this study, we explore fibroblast heterogeneity in keloid, a paradigm of fibrotic skin diseases, by using single-cell RNA-seq. Our results indicate that keloid fibroblasts can be divided into 4 subpopulations: secretory-papillary, secretory-reticular, mesenchymal and pro-inflammatory. Interestingly, the percentage of mesenchymal fibroblast subpopulation is significantly increased in keloid compared to normal scar. Functional studies indicate that mesenchymal fibroblasts are crucial for collagen overexpression in keloid. Increased mesenchymal fibroblast subpopulation is also found in another fibrotic skin disease, scleroderma, suggesting this is a broad mechanism for skin fibrosis. These findings will help us better understand skin fibrotic pathogenesis, and provide potential targets for fibrotic disease therapies.

[1] Dermatology Hospital, Southern Medical University, Guangzhou, China. [2] Department of Bioinformatics, School of Basic Medical Sciences, Southern Medical University, Guangzhou, China. [3] Cancer Research Institute, School of Basic Medical Sciences, Southern Medical University, Guangzhou, China. [4] State Key Laboratory of Organ Failure Research, National Clinical Research Center of Kidney Disease, Key Laboratory of Organ Failure Research (Ministry of Education), Guangzhou, China. [5] These authors contributed equally: Cheng-Cheng Deng, Yong-Fei Hu, Ding-Heng Zhu. ✉email: rongzhili@smu.edu.cn; yangbin1@smu.edu.cn

Fibrosis, characterized by fibroblast proliferation and excessive accumulation of extracellular matrix (ECM), contributes to a high level of morbidity and mortality worldwide[1,2]. Fibrosis can affect any organ, leading to progressive tissue scarring and organ dysfunction[1,2]. Fibrotic skin diseases involve the accumulation of ECM components in the dermis, which include scleroderma, hypertrophic scar, keloid, and graft-vs.-host diseases[3,4]. The global impact of fibrotic skin diseases is significant, which affect millions of people worldwide[3,5]. To date, the etiopathogenesis of fibrotic skin diseases has not been thoroughly elucidated, and radical treatments are still lacking.

Many cell types, such as vascular endothelial cells, immune cells, and fibroblasts, that contribute to fibrosis have been identified[1–3]. Fibroblasts are the centric cell type in the process of skin fibrosis, which leads to ECM accumulation and inflammation[3,4,6,7]. Fibroblasts in fibrotic diseases exhibit overwhelming proliferative potential, increased migration, and invasion capacity, and increased ECM deposition, which contribute to fibrosis pathogenesis[4,6]. These actions are primarily driven by fibrogenic growth factors, such as TGFβ, FGF, PDGF, VEGF, and POSTN[4,6,8,9].

For a long time, it was assumed that fibroblasts were a uniform population of spindle-shaped cells. However, emerging evidence indicates that fibroblasts are actually a morphologically and functionally heterogeneous cell population[3,7,10–12]. The development of single-cell RNA-sequencing (scRNA-seq) gives us an opportunity to explore fibroblast heterogeneity of the skin under homeostasis and pathology. scRNA-seq suggested that fibroblasts can be divided into multiple subgroups in normal human dermis[13–16]. scRNA-seq has also been used to study the heterogeneity of fibroblasts in some fibrotic diseases, such as lung fibrosis, systemic sclerosis, and Dupuytren's disease[17–19]. However, to our knowledge, a study regarding scRNA-seq application for exploring fibroblast heterogeneity in fibrotic skin diseases is still absent.

In this study, we performed scRNA-seq analysis in keloid, a paradigm of fibrotic skin diseases[6,20]. Our results suggested that keloid fibroblasts can be divided into 4 subpopulations. Compared to normal scar tissue, the percentage of a subpopulation of fibroblasts expressing mesenchymal cell markers was significantly increased in keloid. Further functional studies revealed that this subgroup of fibroblasts may be responsible for the overexpression of collagens in fibrotic skin diseases through POSTN. These findings will help us more thoroughly understand fibrotic skin diseases and provide potential targets for fibrosis therapies.

## Results

### Single-cell RNA-seq reveals cell heterogeneity of normal scar and fibrotic skin disease dermis tissues.
To dissect the cellular heterogeneity and explore the regulatory changes of fibrotic skin diseases, we performed scRNA-seq on keloid, a paradigm of fibrotic skin diseases, and normal scar dermis tissues (Fig. 1a). We only used the dermis for scRNA-seq analysis because keloid represents a skin dermis fibrotic disease. After stringent quality control (Supplementary Fig. 1a, b), we obtained transcriptomes of 40,655 cells (keloid: 21,488; normal scar: 19,167). Unsupervised Uniform Manifold Approximation and Projection (UMAP)-clustering revealed 21 cell clusters (Fig. 1b and Supplementary Fig. 1c), which were further classified as transcriptional cluster proximity via a phylogenetic cluster tree (Fig. 1c). We identified 5 fibroblast clusters (C2, C4, C8, C14, C15) and 3 endothelial cell clusters (C1, C6, C7), which accounted for the majority of sequenced cells. Some cells expressed keratinocyte cell markers, which resulted from incomplete removal of the epidermis. We were unable to

characterize clusters 18 and 21 with specific marker genes so these clusters were named "unknown". Based on hierarchical clustering (Fig. 1c) and established lineage-specific marker genes (Fig. 1d, e), we assigned these clusters into 9 cell lineages. The fibroblast lineage was identified by COL1A1, and the endothelial lineage was identified by ENG (Fig. 1e).

We next analyzed the proportions of these cell lineages in keloids and normal scars. We removed the cells in the epidermis, including keratinocytes and melanocytes, in the proportion analysis. The cell lineages of keloid and normal scar dermis showed distinct relative cell number ratios (Fig. 1f). Increased proportions were observed for endothelial cells and smooth muscle cells in keloid tissue, which is consistent with reports that keloids exhibit increased angiogenesis[6,20] (Fig. 1f). The proportions of fibroblasts were decreased in keloid tissues compared to normal scar tissues, perhaps resulting from the excessive expansion of endothelial cells and smooth muscle cells. We next explored the number of differentially expressed genes between keloid and normal scar clusters. The results showed that the fibroblast had the largest difference (Fig. 1g), suggesting that fibroblasts undergo significant changes during the fibrotic progress.

### Dermal fibroblasts subcluster into distinct cell populations and mesenchymal fibroblasts are increased in fibrotic skin disease dermis.
Because fibroblasts undergo significant changes during the fibrotic progress in keloid (Fig. 1g), and fibroblasts are important for fibrotic pathogenesis, we next performed unsupervised clustering on all keloid and normal scar fibroblasts and observed further heterogeneity with 13 subclusters, sC1 through sC13 (Fig. 2a). Figure 2b, c shows the cell proportions of the fibroblast subclusters from keloid and normal scar. From the results, we can see that the proportion of sC4 was consistently increased in keloid samples compared to normal scar samples (Fig. 2b, c).

A recent study suggested that normal human dermis fibroblasts can be divided into 4 subpopulations: secretory-papillary, secretory-reticular, mesenchymal, and pro-inflammatory[16]. Secretory-papillary fibroblasts are generally located in the papillary dermis and express known papillary markers, while secretory-reticular fibroblasts are generally located in the reticular dermis and express known reticular markers. Mesenchymal fibroblasts express some mesenchymal progenitor markers, such as COL11A1 and POSTN, which are involved in skeletal system development, ossification, or osteoblast differentiation[16,21,22]. The signatures of pro-inflammatory fibroblasts include inflammatory response, cell chemotaxis, and reduced expression of collagens[16]. Hierarchical cluster analysis suggested that fibroblasts from keloid and normal scar could also be divided into 4 subpopulations (Fig. 2d and Supplementary Fig. 2a–d). The number of cells in sC8 and sC10-13 subpopulations was very small, so we did not include them in further subpopulation analysis. To see whether the 4 fibroblast subpopulations we found could be divided into secretory-papillary, secretory-reticular, mesenchymal, and pro-inflammatory, we used previously identified fibroblast subpopulation markers[16]. The results demonstrated that sC2, sC3, and sC9 were pro-inflammatory fibroblasts, sC6 and sC7 were secretory-papillary fibroblasts, sC1 and sC4 were mesenchymal fibroblasts and sC5 were secretory-reticular fibroblasts (Fig. 2e and Supplementary Fig. 2e).

The expression of specific collagens has been linked to particular fibroblast functions. Therefore, we also analyzed the four fibroblast subpopulations with respect to the level of collagen expression. The analysis revealed that the collagen gene COL11A1 was specifically expressed in sC1 and sC4 fibroblasts (Supplementary Fig. 2f),

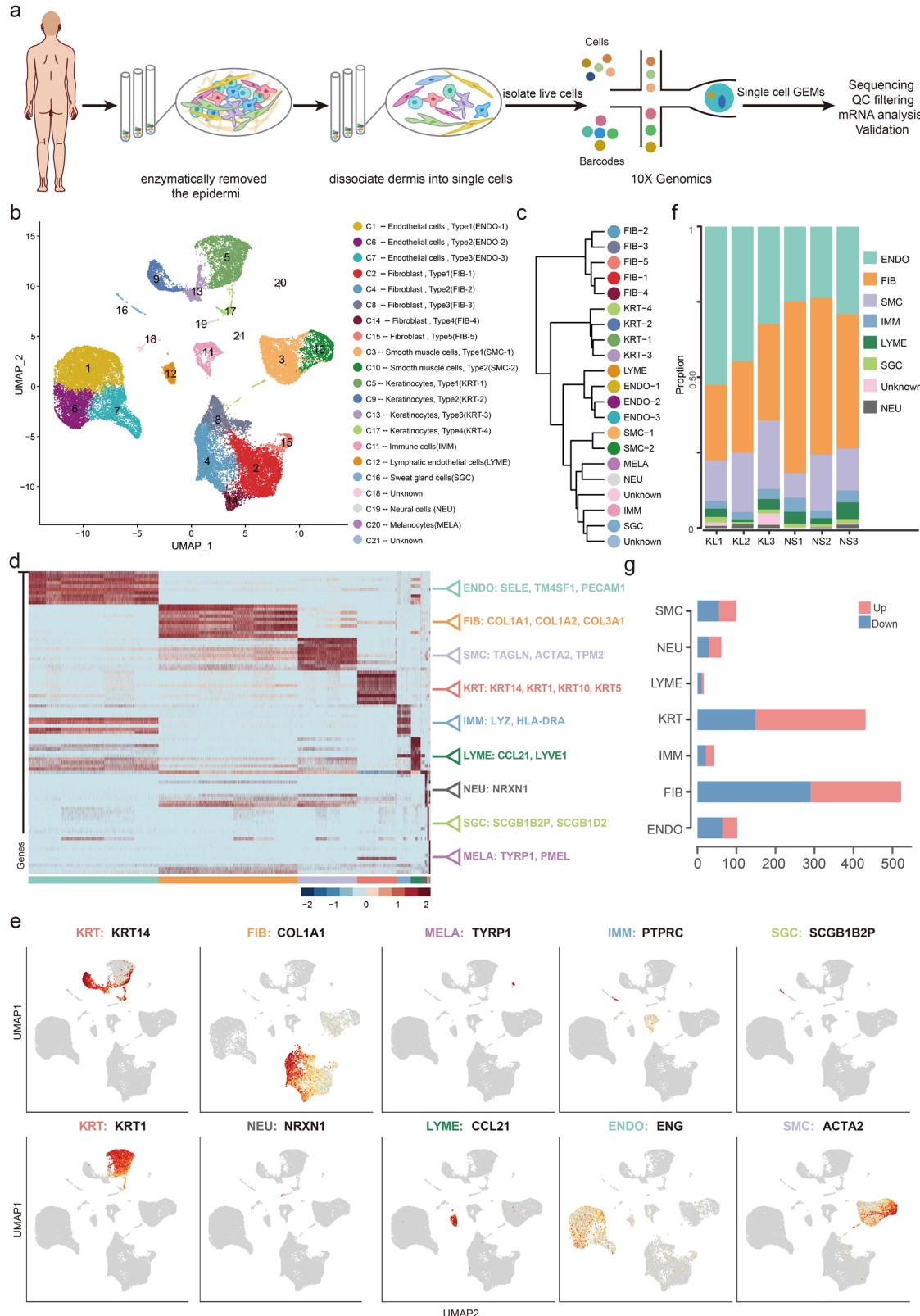

suggesting a stronger mesenchymal component in this cell subpopulation[16,22]. In sC2, sC3, and sC9 pro-inflammatory fibroblasts, the expression of collagens was globally reduced[16] (Supplementary Fig. 2f). In particular, sC6 and sC7 fibroblasts expressed *COL13A1*, *COL18A1*, and *COL23A1*, three known markers of papillary fibroblasts[14,16,23] (Supplementary Fig. 2f). We next compared the proportions of the four fibroblast subpopulations between keloid and normal scar. The results showed that the proportions of secretory-papillary, secretory-reticular, and pro-inflammatory subpopulations were decreased, while the mesenchymal subpopulation was increased in keloid compared to normal scar (Fig. 2f). The increased mesenchymal subpopulation in keloid suggested that this population may be very important for keloid development.

**Fig. 1 Single-cell RNA-seq reveals heterogeneity of normal scar and fibrotic skin disease dermis tissues. a** Illustration of workflow of scRNA-seq in human normal scar and fibrotic skin disease dermis samples. **b** Unbiased clustering of 40655 cells reveals 21 cellular clusters. Clusters are distinguished by different colors. The general identity of each cell cluster is shown on the right. **c** Unsupervised hierarchical clustering of average gene expression showing relatedness of cell clusters (correlation distance metric, average linkage). **d** Heatmap of differentially expressed genes. For each cluster, the top 10 genes and their relative expression levels in all sequenced cells are shown. Selected genes for each cluster are color-coded and shown on the right. **e** Feature plots of expression distribution for selected cluster-specific genes. Expression levels for each cell are color-coded and overlaid onto the UMAP plot. **f** The proportion of cell lineages in keloids (KL) and normal scars (NS). **g** Number of differentially expressed (DE) genes in each cell type with >100 cells available in keloid and normal scars (two-sided Wilcoxon Rank Sum test, Bonferroni correction, log fold change (FC) cutoff of 0.5, and adjusted P-value of <0.05). Red bars indicate upregulated genes, and blue bars indicate downregulated genes in keloid.

We next compared differences between keloid mesenchymal fibroblasts and normal scar mesenchymal fibroblasts. We identified skeletal system development, ossification, and osteoblast differentiation-associated genes, such as COL11A1, COMP, and POSTN, which were significantly increased in keloid mesenchymal fibroblasts (Fig. 2g and Supplementary Data 1). Gene ontology (GO) analysis and Gene Set Enrichment Analysis (GSEA) also suggested that skeletal system development, ossification, and osteoblast associated pathways were enriched in keloid mesenchymal fibroblasts (Fig. 2h, i). These results suggest that not only was the proportion of mesenchymal fibroblasts increased but also the identities of mesenchymal fibroblasts changed in keloid compared to normal scar.

**Characteristics of mesenchymal fibroblasts in fibrotic skin disease dermis.** The scRNA-seq analysis revealed that the proportion of mesenchymal fibroblasts was significantly increased in keloid compared to normal scar (Fig. 2f). Thus, our next work focused on this fibroblast subpopulation. We first explored differentially expressed genes between this subpopulation and other fibroblast subpopulations in keloid. We found that skeletal system development, ossification, and osteoblast differentiation-associated genes such as POSTN and COL11A1, were enriched in the mesenchymal subpopulation (Fig. 3a). The heatmap suggested that gene expression was significantly different between the mesenchymal subpopulation and other subpopulations (Fig. 3b). The increased genes in the mesenchymal subpopulation included some secretory proteins, such as POSTN, COMP, COL11A1, COL12A1, and COL5A2 (Fig. 3c and Supplementary Data 2). We also found that some membrane proteins, such as SDC1, ADAM12, and CD266 (encoded by TNFRSF12A), were increased, while CD9 was decreased, in the mesenchymal subpopulation (Fig. 3c). GO and GSEA analyses suggested that upregulated genes in the mesenchymal subpopulation were associated with the collagen organization process, wound healing, skeletal system development, osteoblast differentiation, and so on (Fig. 3d and Supplementary Fig. 3a).

The lineages, identities, and roles of cells have been reported decided by master transcription factors (TFs)[24,25]. The algorithm for the reconstruction of accurate cellular networks (ARACNe) is a powerful algorithm that can identify master players, especially master TFs, in a gene regulatory network based on a large set of gene expression data[26]. To explore master TFs of the fibroblast subpopulations, we performed ARACNe analysis on our scRNA-seq data (Fig. 3e and Supplementary Fig. 3b–e). Some osteogenesis, chondrogenesis, and ligament and tendon differentiation-associated TFs, such as SCX, CREB3L1, and RUNX2, were enriched in the mesenchymal fibroblast subpopulation (Fig. 3e), consistent with its mesenchymal characteristics.

To validate the findings identified by scRNA-seq, we performed immunofluorescence (IF) staining on skin tissues derived from normal control and keloid. Mesenchymal fibroblasts were identified based on ADAM12 and NREP expression (Fig. 3c). IF staining results showed that the proportion of ADAM12+/NREP+ cells was higher in keloids than in normal control (Fig. 3f, g).

These staining results validated the results of scRNA-seq. ADAM12+/NREP+ cells showed a scattered presence in the dermis, and were not enriched in the upper dermis or lower dermis. They were also not enriched around certain skin structures such as hair follicles or blood vessels.

Myofibroblasts have been reported to be increased and to be essential cells for extracellular matrix production in fibrotic diseases[3,6]. To check the relationship between myofibroblasts and the mesenchymal fibroblasts, we analyzed the expression of ACTA2, a marker of myofibroblasts[3], in our single-cell data. We found that the number of myofibroblasts was increased in keloids compared to normal scars (26.0% ± 4.3% vs 13.3% ± 6.4%) (Supplementary Fig. 4a). In keloids, myofibroblasts were enriched in the mesenchymal fibroblast subpopulation (53.8% ± 9.2%) and existed in the other three fibroblast subpopulations (pro-inflammatory fibroblast: 29.7% ± 10.6%; secretory-papillary fibroblast: 8.9% ± 1.2%; secretory-reticular fibroblast: 7.6% ± 0.3%). Only part of mesenchymal fibroblasts was positive for ACTA2 expression (36.6% ± 8.0%) (Supplementary Fig. 4a). We also analyzed the expression of ADAM12 and α-SMA (encoded by ACTA2) in keloid and normal scar tissues by immunofluorescence. The immunofluorescence experiments showed similar results that only part of ADAM12 positive mesenchymal fibroblasts were α-SMA positive in keloid (Supplementary Fig. 4b). These results suggested that part of mesenchymal fibroblasts were myofibroblasts, and most of the myofibroblasts were in the mesenchymal fibroblast subpopulation in keloid.

**Potential ligand–receptor interactions analyses in mesenchymal fibroblast subpopulations.** The single-cell dataset provided us with a unique chance to analyze cell–cell communication mediated by ligand–receptor interactions. To define the cell–cell communication landscape between fibroblast subpopulations and other cells in keloid and normal scar, we performed analysis using CellPhoneDB 2.0[27], which contains a repository of ligand–receptor interactions and a statistical framework for predicting enriched interactions between two cell types from single-cell transcriptomics data. We observed a dense communication network among fibroblasts and other cells in both normal scar and keloid (Fig. 4a, b). Under both conditions, the most abundant interactions occurred among the four fibroblast subpopulations, suggesting the importance of fibroblasts interaction signaling in the dermal. In normal scar, interactions between secretory-reticular fibroblasts and other cells were most abundant (Fig. 4a), but in keloid, interactions between mesenchymal fibroblasts and other cells were most abundant (Fig. 4b), suggesting the important role of mesenchymal fibroblasts in keloid development.

We next identified ligand–receptor pairs between other cells and fibroblast subpopulations in keloid versus normal scar (fibroblast subpopulations express receptors and receive ligand signals from other cells). Significantly altered signals included some fibrosis-related signals, such as TGFβ1 signaling, VEGF

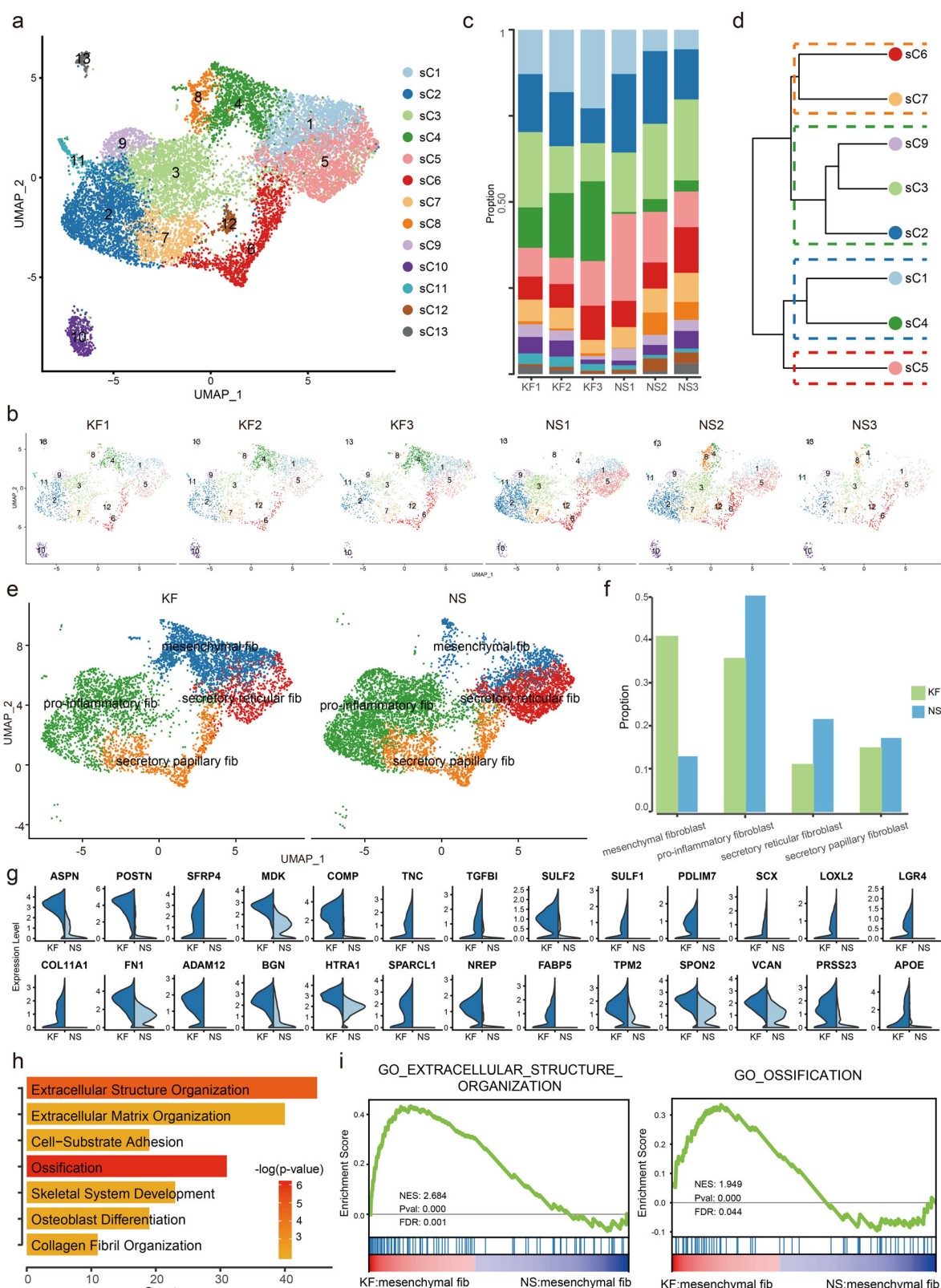

signaling, and POSTN signaling (Fig. 4c, left panel). Among them, TGFβ1 signaling was most significantly altered. The interactions between TGFβ1 and its receptors, TGFβR1 and TGFβR2, were markedly increased in keloid compared to normal scar (Fig. 4c, left panel). Notably, POSTN signaling was only altered in mesenchymal fibroblast subpopulations (Fig. 4c, left panel). In addition, we explored alterations in ligand signals

broadcasted by fibroblasts (Fig. 4c, right panel). We found that fibroblasts may affect other cells in the keloid through alterations in ligand–receptor interactions of TGFβ1 signaling. We also found significantly increased NOTCH signaling, such as JAG1-NOTCH1, in keloid (Fig. 4c, right panel). Fibrosis inhibition associated FGF2 ligand–receptor interactions were significantly decreased in keloid, which was consistent with previous

**Fig. 2 Keloid and normal scar fibroblasts subcluster into distinct cell populations. a, b** Subclustering of keloid and normal scar fibroblasts (cells from clusters C2, C4, C8, C14, and C15 shown in Fig. 1) further identified 13 distinct subtypes. Color-coded UMAP plot is shown and each fibroblast subcluster (sC1 through sC13) is defined on the right. KF: keloid fibroblasts, NS: normal scar fibroblasts. **c** Cell proportions of fibroblast subclusters in keloids and normal scars. Cells of sC4 were significantly increased in keloid samples compared to normal scar samples. **d** Unsupervised hierarchical clustering showing relatedness of fibroblast subclusters (Euclidean distance metric, average linkage). **e** Keloid and normal scar fibroblasts could be divided into 4 subpopulations: secretory-papillary, secretory-reticular, mesenchymal, and pro-inflammatory. **f** The proportions of 4 fibroblast subpopulations in keloid and normal scar. **g** Violin plots showing representative differentially expressed genes between keloid mesenchymal fibroblasts and normal scar mesenchymal fibroblasts. **h** GO Biological Process enrichment analysis of differentially expressed genes in mesenchymal fibroblasts between keloid and normal scars. **i** GSEA enrichment plots for representative signaling pathways upregulated in keloid mesenchymal fibroblasts compared to normal scars.

reports[28,29] (Fig. 4c, right panel). Representative ligand–receptor circle figures also indicated that fibrosis signaling interactions, such as TGFβ, POSTN, and PDGF, were significantly increased in keloid compared to normal scar (Fig. 4d, e and Supplementary Fig. 5).

**Mesenchymal fibroblasts promote the expression of collagens in the keloid partially through POSTN.** To explore the characteristics and function of mesenchymal fibroblasts in fibrosis, we developed a strategy to isolate these cells. Based on scRNA-seq data, all of the fibroblasts in keloid expressed CD90, a well-known fibroblast marker[14]. Most mesenchymal fibroblasts were CD266 (encoded by TNFRSF12A) positive and CD9 negative compared to other fibroblasts (Fig. 3d). Therefore, we flow-sorted keloid fibroblasts that were CD90+ and CD266+/CD9− or CD90+ other cells (other fibroblasts) (Fig. 5a and Supplementary Fig. 6). qRT-PCR and western blot validated that mesenchymal fibroblasts marker genes were significantly enriched in CD266+/CD9− fibroblasts compared to other fibroblasts (Fig. 5b and c).

To examine the characteristics of CD266+/CD9− in depth, we performed RNA-seq to compare the gene expression of CD266+/CD9− fibroblasts and other fibroblasts. The results showed that POSTN, ASPN, COMP, and COL11A1 were increased in the CD266+/CD9− group (Supplementary Data 3), which was consistent with the scRNA-seq results. GO and GSEA analyses showed that upregulated genes in CD266+/CD9− fibroblasts were associated with extracellular matrix organization, collagen fibril organization, skeletal system development, chondrocyte development, and so on (Fig. 5d, e), which was also consistent with the scRNA-seq results. Taken together, these results indicate that CD266+/CD9− fibroblasts had the mesenchymal fibroblasts identity.

POSTN, which encodes periostin, plays an important role in keloid formation and increases collagen expression in keloid[8,30,31]. Increased collagen expression is an important characteristic of skin fibrosis. Considering that one of the primary features of mesenchymal fibroblasts in keloid was abnormal ECM proteins expression including POSTN, and that POSTN-ITGAV;ITGB5 interactions between mesenchymal and other fibroblasts were significantly increased in keloid compared to normal scar, we next explored the function of mesenchymal fibroblasts on keloid fibroblast collagen expression. We first flow-sorted CD266+/CD9− fibroblasts or other fibroblasts and cultured them in dishes. We next collected the supernatant of CD266+/CD9− or other fibroblasts to treat other fibroblasts. We found that the expression of collagen I and III was higher in the CD266+/CD9− supernatant treatment group than that of other fibroblasts group (Fig. 5f, g). To see whether the increased expression of collagen I and III resulted from POSTN, we next mixed the CD266+/CD9− supernatant with a POSTN neutralizing antibody, OC-20, which blocks POSTN and integrins interactions[32,33]. OC-20 inhibited the increased expression of collagen I and III in fibroblasts of

CD266+/CD9− supernatant treatment significantly (Fig. 5h), suggesting that the mesenchymal fibroblasts promoted collagen synthesis of the other fibroblasts in keloid partially through POSTN.

**sC4 fibroblasts are more mesenchymal-like than sC1 fibroblasts.** The mesenchymal fibroblast subpopulation included sC1 and sC4 subgroups. To further explore the relationships of the mesenchymal fibroblast subgroups in keloid, we performed diffusion-pseudotime (DPT) analysis of sC1 and sC4 fibroblasts (Fig. 6a, b). Ordering of sC1 and sC4 cells in pseudotime arranged them into a major trajectory. Fibroblasts from sC1 and sC4 were distributed across the pseudotime space, with sC1 cells primarily occupying the lower space of the major trajectory, and sC4 cells primarily occupying the upper space (Fig. 6a). Furthermore, mesenchymal progenitor marker genes such as POSTN and COL11A1 primarily occupied the upper space (Fig. 6c and Supplementary Fig. 7a).

We next performed RNA velocity analysis, which considers both spliced and unspliced mRNA counts to predict potential directionality and speed of cell state transitions. This analysis distinguished about four sets of vectors, defined as Paths. From the Paths, we can see a branched trajectory with two major branches, i.e., Path2 and Path3,4, and a "pre-branch" Path1, which represents the initial states of fibroblasts (Fig. 6d). sC4 fibroblasts constituted the vast majority of the "pre-branch" and Path2, and sC1 fibroblasts constituted the majority of Path3 and 4 (Fig. 6d and Supplementary Fig. 7b). These analyses suggested that sC4 fibroblasts may be lower differentiated than sC1 fibroblasts. GO analyses suggested that collagen fibril organization, bone development, ossification, and so on were enriched in Path2, and some pathways involved in immune response and wound healing were enriched in Path3,4 (Supplementary Fig. 7c–e). We next analyzed expression trends along trajectories for some transcription factors in Path2 and Path3,4. The expression of SOX4 and JUN maintained high levels in Path2, and the expression of ID2 and CARHSP1 maintained high levels in Path3,4 (Supplementary Fig. 7f).

Pseudotime and RNA velocity analyses suggested that sC4 fibroblasts may be lower differentiated than sC1 fibroblasts. Interestingly, the proportion of sC4 fibroblasts was significantly increased in keloid compared to normal scar. Thus, we next focused on sC4 fibroblasts. Although both sC1 and sC4 fibroblasts are mesenchymal fibroblasts, their gene expression exhibited some differences. Mesenchymal progenitor-associated genes, such as POSTN, ADAM12, and COL11A1, were higher in sC4 fibroblasts than in sC1 fibroblasts (Fig. 6e and Supplementary Data 4). GO and GSEA analyses also suggested that extracellular matrix organization, ossification, chondrocyte development, and so on were enriched in sC4 fibroblasts compared to sC1 fibroblasts (Fig. 6f, g). Taken together, these results indicate that sC4 fibroblasts are more mesenchymal-like, and maybe more important in keloid development.

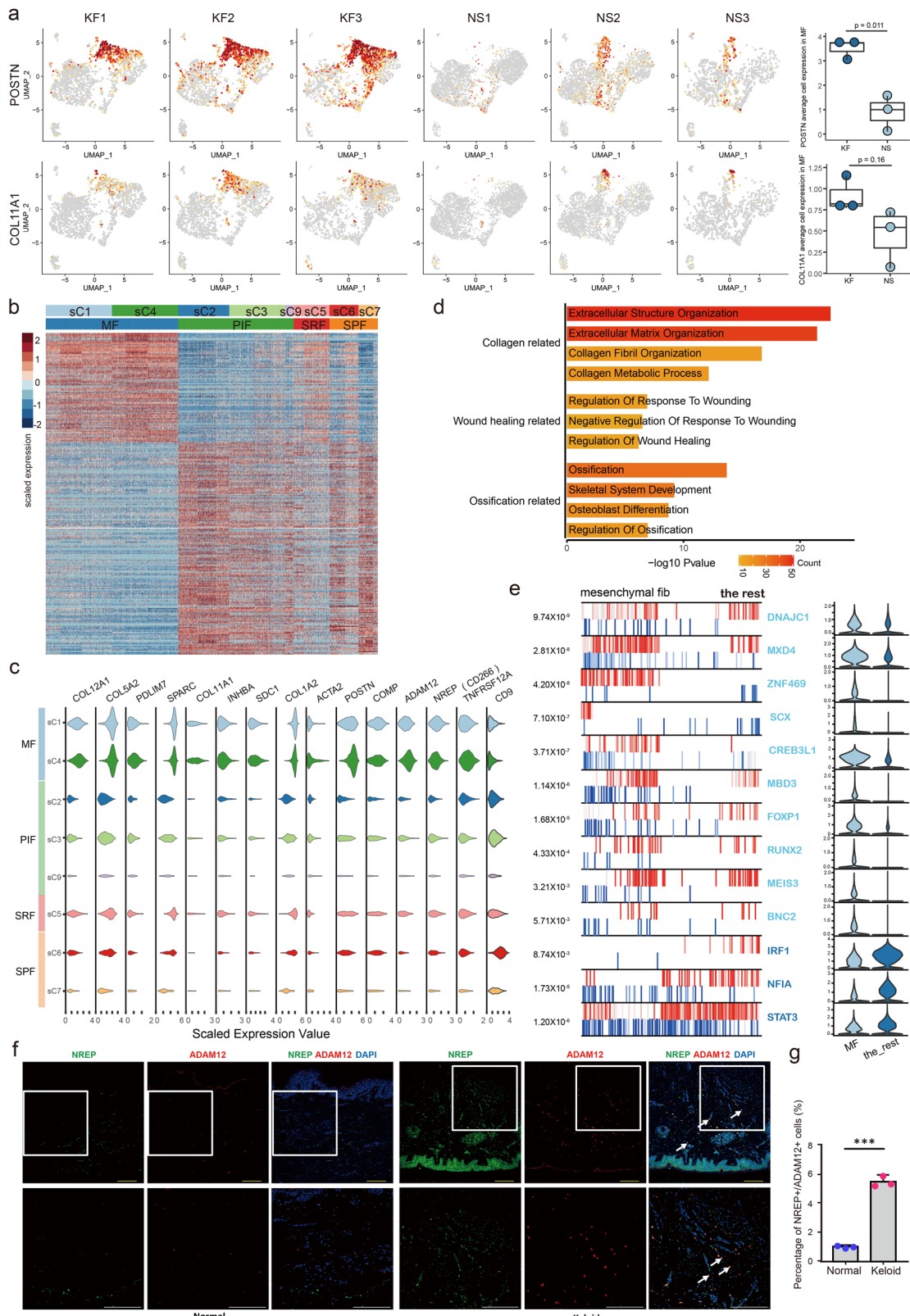

**Mesenchymal fibroblasts are increased in scleroderma**. To examine the consistency of our findings in other fibrotic skin diseases, we investigated scRNA-seq data from a recent scleroderma research[34]. We compared fibroblast data of scleroderma with our No. 3 normal scar fibroblast data because they had similar fibroblasts numbers. We performed unsupervised clustering on all scleroderma and normal scar fibroblasts and

observed heterogeneity with 9 subclusters (Fig. 7a, b). We found that cluster 7 was increased in scleroderma compared to normal scar (Fig. 7c). Mesenchymal fibroblast markers POSTN, ADAM12, COMP, and NREP were extensively expressed in scleroderma cluster 7 fibroblasts (Fig. 7d). Correlation analysis also suggested that scleroderma cluster 7 fibroblasts were most analogous to mesenchymal fibroblasts in keloid (Fig. 7e).

**Fig. 3 Characteristics of mesenchymal fibroblasts in fibrotic skin disease. a** Feature plots of the expression distribution for POSTN and COL11A1 in keloid fibroblasts (KF) and normal scar fibroblasts (NS). Expression levels for each cell are color-coded and overlaid onto the UMAP plot. The average cell expression of each gene in mesenchymal fibroblasts between KF and NS were shown in the right panel (Two-sided unpaired t-test). $n = 3$ biologically independent samples. In box plots, lines in the middle of boxes correspond to median values. Lower and upper hinges correspond to the first and third quartiles, and the upper whisker extends from the hinge to the largest value no further than $1.5 \times$ IQR (inter-quartile range) from the hinge. The lower whisker extends from the hinge to the smallest value at most $1.5 \times$ IQR of the hinge. **b** Heatmap of differentially expressed genes in each fibroblast subpopulations. MF: mesenchymal fibroblast, PIF: pro-inflammatory fibroblast, SRF: secretory-reticular fibroblast, SPF: secretory-papillary fibroblast. **c** Violin plots illustrating the expression of some marker genes in each fibroblast subpopulations. **d** Functional enrichment of upregulated genes in mesenchymal fibroblasts [Fisher exact-test, corrected for multiple comparisons using Benjamini and Hochberg method, adjusted P-value of <0.05]. **e** The top 13 candidate master regulators for mesenchymal fibroblasts identified by an algorithm for master regulator analysis algorithm (MARINa). Violin plots showing the relative expression levels of each master regulator in the right panel. **f** Immunofluorescence staining of NREP and ADAM12 in normal and keloid tissues. Lower panels are the insets of upper panels. Arrowheads indicate NREP$^+$/ADAM12$^+$ cells. Scale bar = 200 μm. **g** Percentage of NREP$^+$/ADAM12$^+$ cells in normal and keloid tissues. Data are presented as mean values ± SD ($n = 5$ images examined over 3 independent experiments). Two-sided unpaired t-test, ***$P = 0.00013$.

Immunofluorescence staining results showed that the proportion of ADAM12$^+$/NREP$^+$ cells was higher in scleroderma tissue than in normal control tissue (Fig. 7f, g). Taken together, these results indicated that increasing mesenchymal fibroblasts may be a universal mechanism in fibrotic skin diseases.

## Discussion

Although fibrotic skin diseases have been extensively studied, key mechanisms leading to the development of these diseases are still not well understood[2,4]. In addition, treatments to prevent or treat skin fibrosis are scarce and not effective[2,4]. Fibrotic skin tissues include multiple cell subpopulations with diverse genetic and phenotypic characteristics. How this heterogeneity emerges in developing fibrosis remains unclear[3]. Herein, we built a single-cell atlas of a representative human fibrotic skin disease, keloid, and explored the characteristics and key regulatory pathways of distinct fibroblast subtypes. These findings will help us understand skin fibrotic pathogenesis in depth, and provide potential targets for clinical therapies of these diseases.

Fibroblasts are increasingly recognized as central mediators of diverse fibrotic diseases, and here, we identified 13 fibroblast subgroups in human fibrotic skin disease tissues by using scRNA-seq (Fig. 2a, b). Further cluster analysis suggested that these fibroblast subgroups could be divided into 4 subpopulations: secretory-papillary, secretory-reticular, mesenchymal, and pro-inflammatory (Fig. 2e and Supplementary Fig. 2), which was consistent with previous findings in normal human skin[16]. The percentage of cells in the mesenchymal subpopulation was significantly increased in keloid compared to normal scar tissues (Fig. 2e, f). Significantly, we also found increasing mesenchymal fibroblast subpopulation in scleroderma (Fig. 7c–g), another fibrotic skin disease, suggesting that this may be a universal mechanism in skin fibrosis. According to our results and the results from a previous study[16], these mesenchymal fibroblasts expressed genes associated with skeletal system development, ossification, and osteoblast differentiation (Fig. 3c, d), suggesting a stronger mesenchymal component in this cell subpopulation. Some skeletal system development, tendon development, and osteoblast differentiation-associated master TFs, such as SCX and RUNX2[35,36], were also enriched in this subpopulation (Fig. 3e). These TFs may play an important role in deciding the cell identities of these fibroblast subpopulations. Increases of these mesenchymal fibroblasts in keloid indicated that skin fibrosis may be associated with osteogenesis and chondrogenesis.

One of the characteristics of the mesenchymal fibroblasts in keloid is the high expression of secretory proteins such as POSTN, COMP, COL11A1, ASPN, and COL5A2 (Fig. 3c and Supplementary Data 2). Previous studies suggested that some of these proteins such as POSTN were increased and could promote

collagens production in keloid[31,37]. However, these studies did not explore which cells these proteins were derived from. We also did not know whether the ECM proteins increase in all cells or only in some cells. Our results indicated that these proteins were primarily expressed in fibroblasts of keloid, and only some fibroblasts expressed these proteins (Fig. 3c). Furthermore, our functional study suggested that the supernatant of this group of fibroblasts increased the expression of collagens in other fibroblasts (Fig. 5f, g). By analyzing single-cell data from scleroderma[34], we also found an increased mesenchymal fibroblast subpopulation highly expressing POSTN, COMP, ASPN, and so on in fibrosis tissues compared to normal tissues (Fig. 7d). These results suggested that the mesenchymal fibroblast subpopulation may have an important role in multiple skin fibrosis diseases, and may serve as target cells for fibrosis treatment.

Myofibroblasts have been reported to be significantly increased and contribute to collagen formation in fibrotic diseases[3,6]. Our results suggested that a small part of mesenchymal fibroblasts were myofibroblasts, and most of the myofibroblasts were in the mesenchymal fibroblast subpopulation in keloid (Supplementary Fig. 4). Our discovery indicates that mesenchymal fibroblasts play an important role in collagen deposition in keloid, and most of the myofibroblasts were in the mesenchymal fibroblast subpopulation and may have the same function as mesenchymal fibroblast. Our discovery is consistent with the previous hypothesis that myofibroblasts contribute to collagen formation in the scars but the difference is that our discovery might expand the myofibroblast hypothesis.

Fibroblasts are known to interact with other cell types in the skin. scRNA-seq provides opportunities to identify communicating pairs of cells based on the expression of cell-surface receptors and their interacting ligands. The results showed that fibroblasts interacted with all cells we identified in our scRNA-seq (Fig. 4a). Notably, in the normal scar, the most abundant interactions occurred between secretory-reticular fibroblasts and other cells. However, in fibrotic skin, the most abundant interactions occurred between mesenchymal fibroblasts and other cells (Fig. 4a). These results suggest that these mesenchymal fibroblasts play an important role in skin fibrosis development, which is consistent with their increases in skin fibrosis. We found a marked increase of TGFβ-TGFβ receptor interactions in keloid compared to normal scar (Fig. 4b), indicating the central roles of the TGFβ pathway in fibrosis development. We also identified some previously reported fibrosis-associated interactions, such as NOTCH and angiogenesis-associated VEGF ligand–receptor interactions (Fig. 4b)[6,38], which were increased in skin fibrosis. Interestingly, we found that the POSTN-ITGAV;ITGB5 interactions between mesenchymal and other fibroblasts were significantly increased in keloid compared to normal scar (Fig. 4b, c). Blocking the interactions by using POSTN

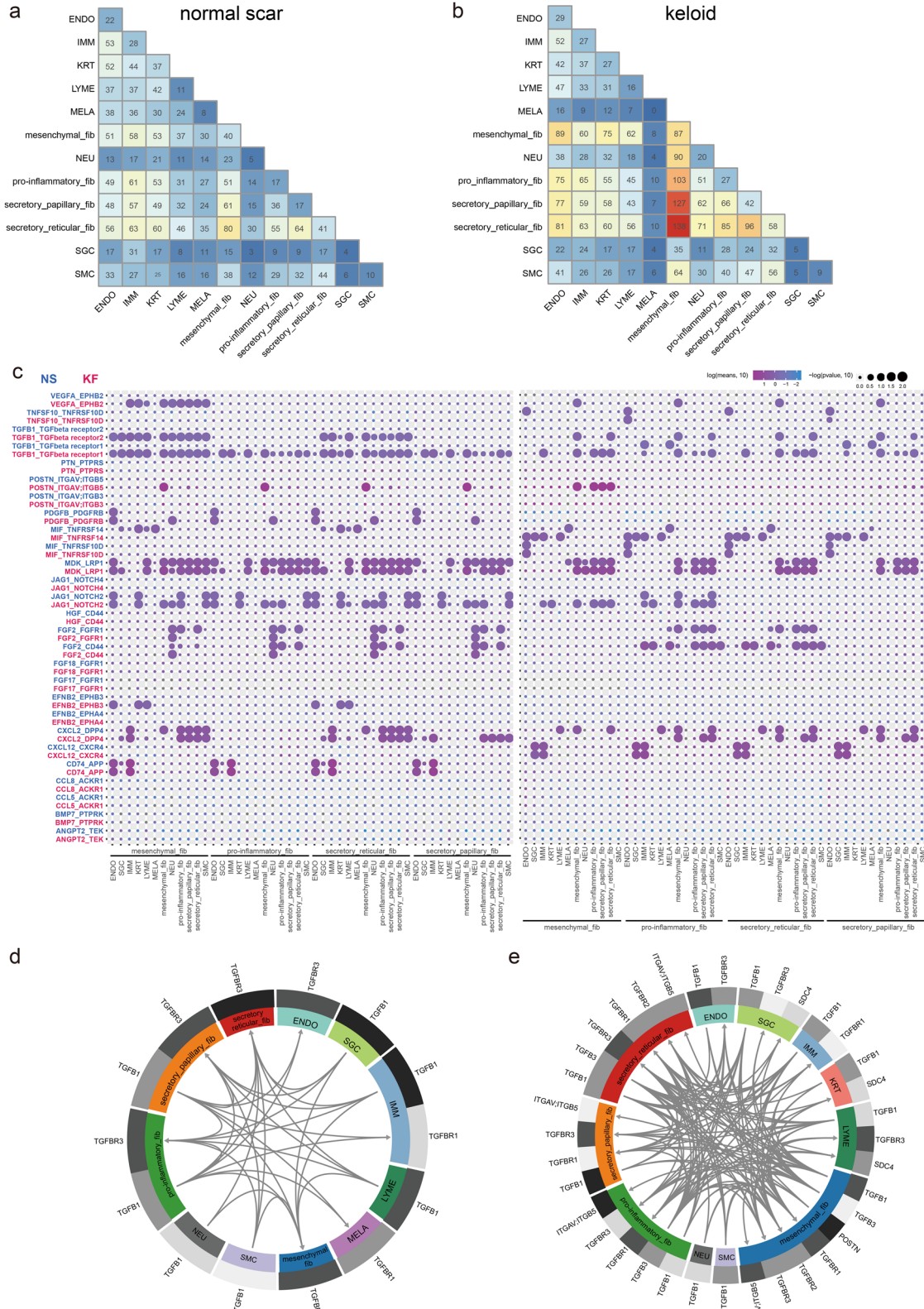

**Fig. 4 Potential ligand–receptor interactions analyses in fibroblast subpopulations. a, b** Heatmap showing the numbers of inter-populations communications with each other in normal scars (**a**) and in keloid tissues (**b**). **c** The ligand–receptor pairs exhibit significant changes in specificity between any one of the population and any one type of fibroblast in normal scars versus keloid. The left panel shows that one type of fibroblast expresses receptors and receives ligand signals from other populations. The right panel shows that one other population expresses receptors and receives ligand signals from one type of fibroblast. **d, e** Putative TGFβ and POSTN relative signaling within fibroblasts and other cell populations in normal scars (**d**) and in keloid tissues (**e**). All arrows are pointing to the receptors. Average expression levels for each ligand/receptor are presented as a heatmap plot. Black indicates maximum relative expression and white indicates low or no expression of a particular gene.

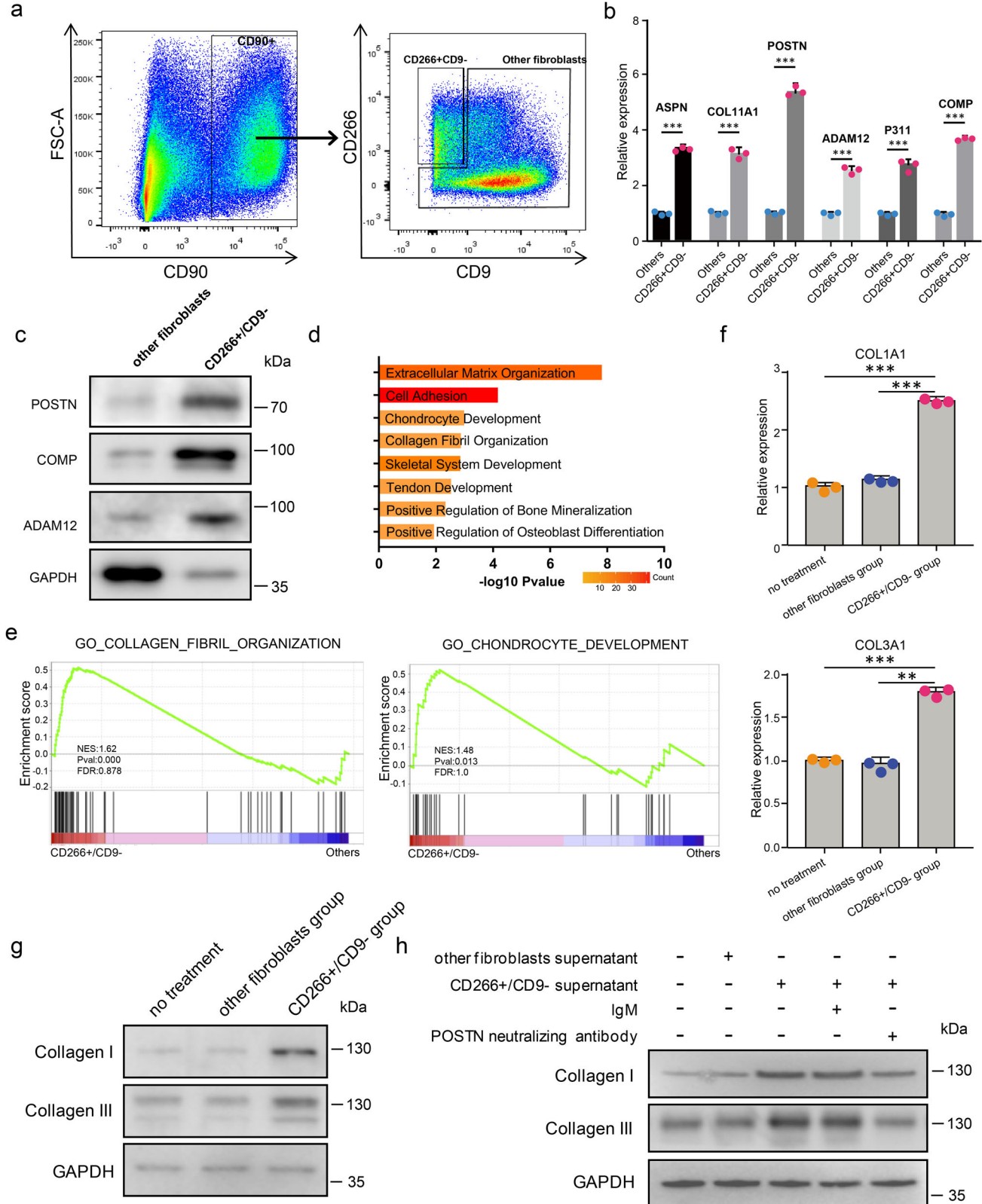

neutralizing antibodies inhibited the increase of collagens in other fibroblasts treated with mesenchymal fibroblast supernatant (Fig. 5h), indicating that these interactions are important for keloid collagen overexpression and may serve as therapeutic targets.

One of the most interesting findings in our study was that a fibroblast subgroup, sC4, was significantly increased in fibrotic skin compared to normal scar (Fig. 2c), indicating its important

role in skin fibrosis. The sC4 subgroup was almost absent in NS1, and in NS2 and NS3, this subgroup was small (Fig. 2c). Some osteogenesis and chondrogenesis associated secretory proteins, such as POSTN and COL11A1, were higher in sC4 than in sC1 (Fig. 6e and Supplementary Data 4), the other mesenchymal fibroblast subgroup, suggesting that sC4 fibroblasts may be more pluripotent. Pseudotime and RNA velocity

**Fig. 5 The supernatant of mesenchymal fibroblasts promotes collagen expression in keloid fibroblasts. a** Isolation of CD266[+]CD9[−] and other fibroblasts from keloid dermis by flow cytometry. **b** qRT-PCR assay of some mesenchymal fibroblast marker genes expression in CD266[+]/CD9[−] fibroblasts and other fibroblasts from keloid dermis. Data are presented as mean values ± SD ($n = 3$ biologically independent experiments.). Two-sided unpaired $t$-test, ***$P < 0.001$ (ASPN: $P = 0.00003$, COL11A1: $P = 0.00019$, POSTN: $P = 0.00004$, ADAM12: $P = 0.00015$, P311: $P = 0.00019$, COMP: $P = 0.00002$). **c** Western blot assay of mesenchymal fibroblast marker genes expression in CD266[+]/CD9[−] fibroblasts and other fibroblasts from keloid dermis. The experiments were repeated three times with three different fibroblast donors, and here a representative result was shown. **d** GO Biological Process enrichment analysis of differentially expressed genes between keloid CD266[+]/CD9[−] fibroblasts and other fibroblasts. **e** GSEA enrichment plots for representative signaling pathways upregulated in keloid CD266[+]/CD9[−] fibroblasts compared to other fibroblasts. (NES normalized enrichment score, corrected for multiple comparisons using FDR method, $P$-value were showed in plots). **f, g** Keloid other fibroblasts were treated with the supernatant of CD266[+]/CD9[−] or other fibroblasts. The expression of collagen I and collagen III was analyzed by qRT-PCR (**f**) or western blot (**g**). Data are presented as mean values ± SD ($n = 3$ biologically independent experiments). Two-sided unpaired $t$-test, **$P < 0.01$, ***$P < 0.001$. (COL1A1: $P = 0.000009$ and $P = 0.000015$, respectively; COL3A1: $P = 0.00006$ and $P = 0.00214$, respectively). The experiments were repeated three times with three different fibroblast donors, and here a representative result was shown. **h** Keloid other fibroblasts were treated as indicated in the figure. The expression of collagen I and collagen III was analyzed by western blot. The POSTN antibody neutralizing experiments were repeated three times with three different fibroblast donors, and here a representative result was shown.

analyses also suggested that sC4 fibroblasts were at the early stage of differentiation and could differentiate into sC1 fibroblasts (Fig. 6a–d). Because of the similar expression levels of membrane proteins between sC4 and sC1 subgroups (Supplementary Data 4), we can not flow-sorted them and compare their functions in this study. One interesting question is where did the sC4 fibroblast subgroup comes from. These fibroblasts may come from dedifferentiation of the cells in normal skin or from mesenchymal stem cells in the bone marrow. Our next study will explore the origin of these fibroblasts by using transgenic mice and lineage tracing.

Surgical excision remains the mainstay for the treatment of keloid, and multiple adjuvant therapies, such as pulsed-dye laser ablation, radiation therapy, pressure therapy, $CO_2$ laser ablation, and intralesional steroids, have been used[6,39,40]. In most cases, the patients are effectively treated with these non-targeted therapies, but with varying degrees of recurrence[6,39,40]. Our studies indicate that mesenchymal fibroblasts are important for the overexpression of collagens in keloid through POSTN. We may develop some methods, such as using small molecule inhibitors of POSTN, to target mesenchymal fibroblasts. Inhibiting or eliminating mesenchymal fibroblasts before or after non-targeted therapies in keloid patients may improve the therapeutic effect of keloid.

In conclusion, we provided a systematic analysis of fibroblast heterogeneity in fibrotic skin diseases at single-cell resolution. In addition, we identified an increased mesenchymal fibroblast subpopulation in fibrotic skin diseases involved in collagen overexpression. These findings will help to understand skin fibrotic pathogenesis in depth and identify potential targets for fibrotic disease treatment.

## Methods

**Sample preparation and tissue dissociation.** This study was approved by the Medical and Ethics Committees of Dermatology Hospital, Southern Medical University (2019023), and each patient signed an informed consent before enrolling in this study. All the patients in this research were Han nationality. Keloid tissues were harvested during plastic surgery from three patients confirmed to have clinical evidence of keloid (Supplementary Table 1). All the keloids we used in this study were mature. We used all contents of the keloid samples, including the center and the edge of the samples, and mixed them for further analysis. No patient received chemotherapy, radiotherapy, or intralesional steroids treatment prior to surgery. Normal scar tissues were obtained from three patients who underwent elective scar resection surgery (Supplementary Table 1). Keloids and normal scars were diagnosed on the basis of their clinical appearance, history, anatomical location, and pathology. Excised skin was immersed in physiological saline and then immediately transferred to the lab. The skin tissue was washed twice in PBS. After removal of the adipose tissue under the reticular dermis, samples were cut into 5 mm diameter pieces and incubated with dispase II (Sigma) for 2 h at 37 °C. The epidermis was peeled off and discarded and the dermis was minced into small pieces and digested at 37 °C for 2 h using Collagenase IV (YEASEN, China). The

resulting cell suspension was filtered through a 70 μm cell strainer (BD Falcon), and centrifuged at $400 \times g$ for 10 min. The supernatant was removed and the pellet was washed once with PBS at $400 \times g$ for 5 min. The pellet was then resuspended in PBS + 1% FBS for flow cytometry.

**Single-cell cDNA and library preparation.** Single-cell cDNA, library preparation, and 3′-end single-cell RNA-sequencing were performed by Novogene (Beijing, China). For experiments using the 10× Genomics platform, the Chromium Single Cell 3′ Library & Gel Bead Kit v2 (PN-120237), Chromium Single Cell 3′ Chip kit v2 (PN-120236), and Chromium i7 Multiplex Kit (PN-120262) were used according to the manufacturer's instructions in the Chromium Single Cell 3′ Reagents Kits v2 User Guide. The single-cell suspension was washed twice with 1× PBS + 0.04% BSA. The cell number and concentration were confirmed with a TC20™ Automated Cell Counter.

Approximately 8000 cells were immediately subjected to the 10× Genomics Chromium Controller machine for Gel Beads-in-Emulsion (GEM) generation. mRNA was prepared using 10× Genomics Chromium Single Cell 3′ reagent kit (V2 chemistry). During this step, cells were partitioned into the GEMs along with Gel Beads coated with oligos. These oligos provide poly-dT sequences to capture mRNAs released after cell lysis inside the droplets, as well as cell-specific and transcript-specific barcodes (16 bp 10× Barcode and 10 bp Unique Molecular Identifier (UMI), respectively).

After RT-PCR, cDNA was recovered, purified, and amplified to generate sufficient quantities for library preparation. Library quality and concentration were assessed using Agilent Bioanalyzer 2100.

**3′-end single-cell RNA-sequencing.** Libraries were run on the Hiseq X or Novaseq for Illumina PE150 sequencing. Post-processing and quality control were performed by Novogene using the 10× Cell Ranger package (v2.1.0, 10× Genomics). Reads were aligned to GRCh38 reference assembly (v2.2.0, 10× Genomics).

**Single-cell RNA-sequence data processing.** The sequencing reads were examined by quality metrics, and transcripts were mapped to a reference human genome (hg38) and assigned to individual cells of origin according to the cell-specific barcodes, using the Cell Ranger pipeline (10× Genomics). To ensure that PCR amplified transcripts were counted only once, only single UMIs were counted for gene expression level analysis[41]. In this way, cell-gene UMI counting matrices were generated for downstream analyses. From each sample, unwanted variations and low-quality cells were filtered by removing cells with high and low (>6000 and <200) UMI-counts. Meanwhile, to avoid the effects of doublets, cells that were identified as doublets using Doubletdetection (available from https://github.com/JonathanShor/DoubletDetection) were removed from our data.

Gene expression levels for each cell were normalized by total expression, multiplied by a scale factor (10,000), and log-transformed. Batches were then regressed out, and scaled $Z$ scored residuals of the model were used as normalized expression values. We defined the top 2000 most variable genes based on their average expression and dispersion as highly variable genes (HVG). We reduced the dimensionality of the data by performing the principal component analysis (PCA) on HVG. To identify cell subpopulations, clustering was performed on PCA scores using significant PCs assigned by a randomization approach proposed by Chung and Storey[42,43]. For those replicates (NS1, NS2, NS3 and KF1, KF2, KF3), the first 15 PCs were selected for clustering. To cluster cells, a K-nearest neighbor (KNN) graph constructed on a Euclidean distance matrix in PCA space was calculated and then converted to a shared nearest neighbor (SNN) graph, in order to find highly interconnected communities of cells[44]. Cells were then clustered using the Louvain method to maximize modularity[45]. To display data, the Unsupervised Uniform Manifold Approximation and Projection (UMAP) was applied to cell loadings of selected PCs, and the cluster assignments from the graph-based clustering were

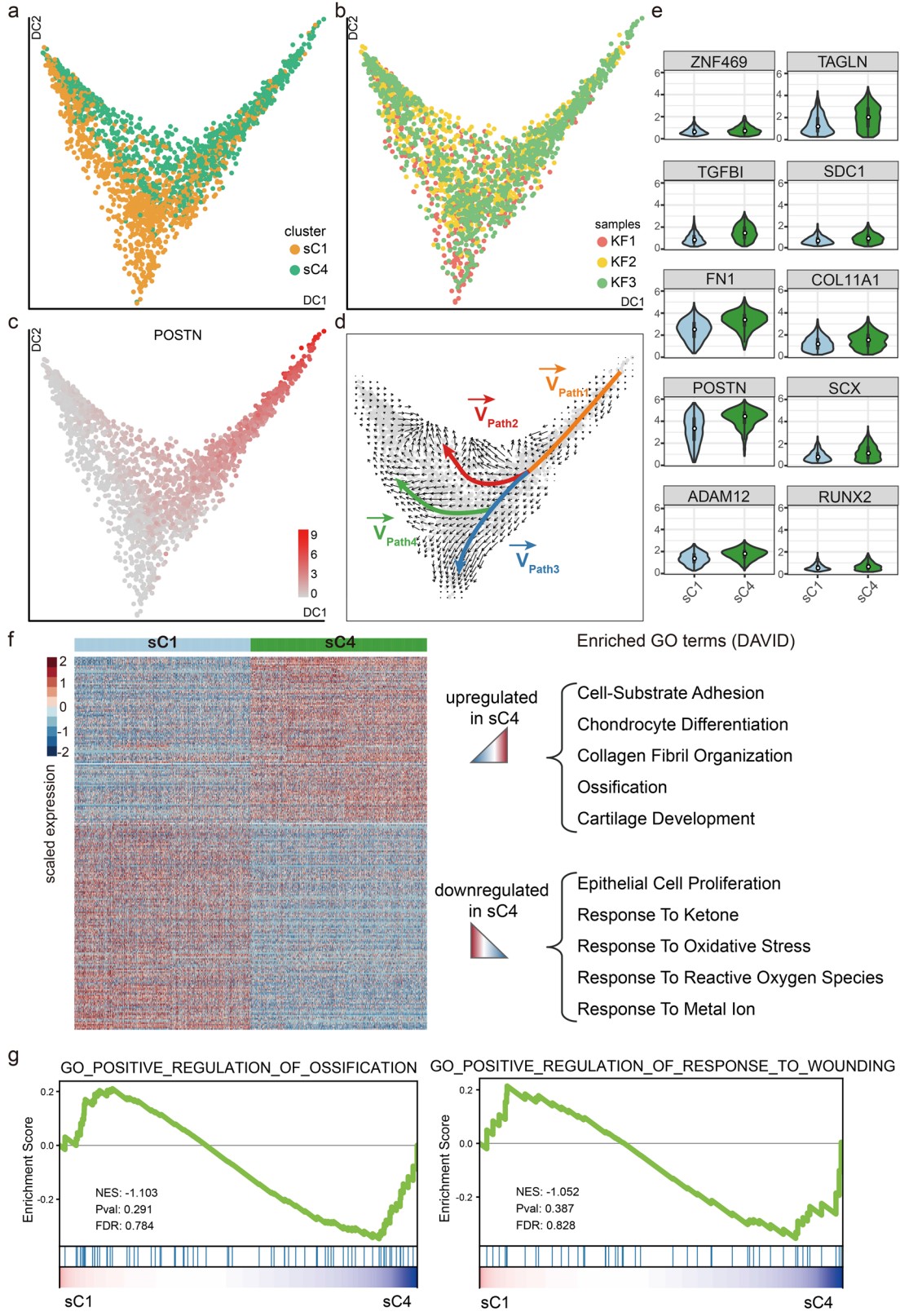

used. For cluster numbers higher than two, cluster-specific marker genes were identified by running the "find_all_markers" Seurat function with parameters logfc. threshold = 0.5 and test.use = "wilcox". To identify differentially expressed genes between two clusters, we used the "find.markers" Seurat function with logfc. threshold = 0.5 and test.use = "wilcox". All analyses described in this section were performed using Seurat R package version 3.0.1.

**Gene sets enrichment analysis**. Gene Ontology[46] functional enrichment (over-representation) of DEGs at $P < 0.05$ was analyzed using a R package clusterProfiler v3.12.0[47]. Gene Set Enrichment Analysis (GSEA) also conducted using GSEA desktop software, and collected gene sets and molecular signatures were obtained from the Molecular Signatures Database[48]. Normalized enrichment scores were acquired using gene set permutations 1000 times, and a cutoff $P$-value of 0.05 was used to filter the significant enrichment results.

**Fig. 6 sC4 fibroblasts are more mesenchymal-like than sC1 fibroblasts. a, b** Results of diffusion-pseudotime analysis of keloid mesenchymal fibroblasts, colored by subcluster (**a**) and by keloid samples (**b**). **c** Diffusion map showing the expression levels of POSTN in keloid mesenchymal fibroblasts. Red indicates maximum relative expression, and gray indicates low or no expression of this gene. **d** RNA velocity analysis distinguished four sets of velocity vectors across the diffusion-pseudotime: Path1 (orange), Path2 (red), Path3 (blue), and Path4 (green). **e** The difference in the expression levels of mesenchymal fibroblasts population-specific genes ($n = 1047$ cells in keloid sC1 cluster, $n = 1048$ cells in keloid sC4 cluster). In embedded box plots, the white dot in the middle of boxes corresponds to median values. Lower and upper hinges correspond to the first and third quartiles, and the upper whisker extends from the hinge to the largest value no further than 1.5 × IQR (inter-quartile range) from the hinge. The lower whisker extends from the hinge to the smallest value at most 1.5 × IQR of the hinge. **f** Heatmap showing differentially expressed genes among the subclusters in mesenchymal fibroblasts, and functional annotations of those dysregulated genes are shown in the right panel. **g** GSEA enrichment plots for representative signaling pathways upregulated in the sC4 subcluster (NES normalized enrichment score, corrected for multiple comparisons using FDR method, P-value were showed in plots).

**Master transcriptional regulator analysis**. ARACNe is widely used to accurately reconstruct gene regulatory networks[49,50]. To infer the transcription factor regulatory network of this study, we used all 1665 human transcription factors of Animal TFDB 3.0[51]. We first performed regulatory network analysis for four types of fibroblasts separately, each with corresponding expression data including one type of fibroblasts and the rest of fibroblasts, using ARACNe-AP software[52]. Second, we performed master regulator analysis using the ssmarina package (deposited in https://figshare.com/articles/dataset/ssmarina_R_system_package/785718), a modification of the MARINa algorithm[53]. Enrichment of the predicted targets was assessed by comparing the gene expression between two groups. Regulators with FDR-corrected P-values below 0.01 were inferred as candidate master regulators between two given groups.

**Cell–cell communication analysis**. To identify potential interactions between and within fibroblasts and other dermal cell populations, we used CellPhoneDB 2.0 with parameters threshold = 0.25 and iterations = 1000[27], which contains a curated repository of ligand–receptor interactions and a statistical framework for inferring lineage-specific interactions. Custom R scripts and circos software were used for analyses and to draw the interaction diagrams[54].

**Pseudotime analysis and RNA velocity analysis**. Diffusion-pseudotime (DPT) analysis was implemented, and diffusion maps were generated using the destiny R package[55]. The number of nearest neighbors, $k$, was set to 100. Velocyto can estimate the RNA velocities of single cells by distinguishing unspliced and spliced mRNAs in standard single-cell RNA-sequencing data. We performed this analysis on fibroblast cells as described by La Manno et al.[56]. Based on velocyto pipeline, annotation of spliced and unspliced reads was performed using the Python script velocyto.py on the Cell Ranger output folder, then merged all keloid data sets, and remain fibroblasts based on the results of our previous cell clustering analysis. PCA analysis was performed with Pagoda2[57] with spliced expression matrix as input, and cell-to-cell distance matrix was calculated using Euclidean distance based on the top 20 principal components. RNA velocity was estimated using a gene-relative model with $k$-nearest neighbor cell pooling ($k = 100$). Velocity fields were projected onto the pseudotime space produced by DPT. Arrows were plotted on an absolute scale. The pseudotime-dependent genes were detected by differentialGeneTest function with fullModelFormulaStr = "~sm.ns(Pseudotime)" in Monocle2[58], the statistically significant threshold was set to a $q$-value < 0.01.

**Integration public data sets**. All Seurat objects for our dataset's individual samples and the public data sets were processed through similar steps as described above to generate a single combined object. Next, combined object was used to perform canonical correlation analysis (CCA) between our dataset and public data sets[59]. Then, CCA subspaces were aligned using 1:30 CCA dimensions, which was followed by integrated UMAP visualization for all cells.

**Immunofluorescence staining**. Immunofluorescence staining was performed on formalin-fixed paraffin-embedded human keloid and normal scar biopsies. Tissue sections were deparaffinized and rehydrated followed by heat-induced antigen retrieval in citrate buffer at pH 6.0 for ~10 min. Antibodies were applied, including mouse anti-ADAM12 (sc293225, Santa Cruz) (1:50), rabbit anti-NREP (bs-0427R, Bioss) (1:100), or rabbit anti-SMA (ab124964, Abcam) (1:300) were incubated overnight at 4 °C. Sections were washed with PBS 3 times, and then labeled with Alexa Fluor 488 (A-11029, ThermoFisher) and 555 (A32732, ThermoFisher) labeled secondary antibodies (1:5000). Slides were coverslipped, using DAPI containing aqueous mounting medium. Images were obtained using a Nikon A1 + confocal laser-scanning microscope.

**Flow cytometry**. Disaggregated dermal cells were labeled with antibodies in PBS + 1% FBS for 30 min at 4 °C. Antibodies were applied, including PE anti-human CD90 (328109, Biolegend) (1:200), FITC anti-human CD9 Antibody (312103, Biolegend) (1:100) and APC anti-human CD266 Antibody (314107, Biolegend)

(1:100). DAPI was used to exclude dead cells. Following incubation, cells were centrifuged at 400 × $g$ for 5 min at 4 °C, and washed three times in PBS + 1% FBS. Pellets were resuspended in PBS + 1% FBS and filtered through a 50 μm cell strainer. Cell sorting was performed on a BD FACSAria™ III Fusion cell sorters. For gate setting and compensation, unlabeled, single-labeled cells and compensation beads (BD) were used as controls. Data analysis was performed using FlowJo software.

**Real-time quantitative PCR**. Real-time quantitative PCR was performed as previously described[60]. Briefly, Total RNA was isolated with TRIzol reagent (Invitrogen). First-strand cDNA was synthesized using a PrimeScript RT Reagent Kit (Takara, Japan). Gene-specific primer pairs were designed by Primer Premier 5.0 software (Supplementary Table 2). Quantitative real-time PCR was performed using SYBR Green Master (Takara, Japan) according to the manufacturer's protocol.

**Western blot**. Western blot was performed as previously described[60]. Briefly, cells were lysed in 1×SDS-PAGE loading buffer (50 mM Tris-HCl at pH 6.8, 2% (W/V) SDS, 0.1% (W/V) BPB, 10% (V/V) glycerol) containing 50 mM β-glycerophosphate, and the lysates were resolved by SDS-PAGE and transferred to polyvinylidene difluoride membranes for western blot using enhanced chemiluminescence detection reagents (Merck Millipore, Billerica, MA). Antibody recognizing collagen I (ab34710) (1:1000) and collagen III (ab7778) (1:1000) were purchased from Abcam. The antibody recognizing GAPDH (60004-1-Ig) (1:5000) was purchased from Proteintech.

**CD266⁺CD9⁻ and the other fibroblasts supernatant treatment experiments**. Flow sorted CD266⁺CD9⁻ and other fibroblasts were maintained in DMEM (Invitrogen) with 10% FBS (Invitrogen) at 37 °C and 5% $CO_2$. After reaching about 80% confluence, the culture mediums were changed to FBS-free DMEM. 24 h later, the supernatants of CD266⁺CD9⁻ and other fibroblasts were collected to treat the other fibroblasts of keloid as indicated in the figures. After 48 h treatment, the RNA and proteins of the cells were collected for qRT-PCR and western blot assays. The POSTN neutralizing antibody OC-20 (2 μg/ml) and control lgM antibody (2 μg/ml) were purchased from AdipoGen (AG-20B-6000PF for OC-20 and ANC-290-810 for lgM antibody), and the OC-20 neutralizing experiments were repeated three times with three different fibroblast donors.

**Statistical analysis**. All experiments were repeated at least three times. Statistical analyses were performed using SPSS software, version 19.0. Data represent mean ± standard deviation. A two-tailed, unpaired Student $t$-test or the Mann–Whitney $U$ test was employed to compare the values between subgroups for quantitative data. $P < 0.05$ was considered to be statistically significant.

**Reporting summary**. Further information on research design is available in the Nature Research Reporting Summary linked to this article.

## Data availability
RNA-seq and scRNA-seq data have been deposited in the Gene Expression Omnibus (GEO) database under accession codes "GSE175866" and "GSE163973", respectively. All other relevant data supporting the key findings of this study are available within the article and its Supplementary Information files or from the corresponding author upon reasonable request. A reporting summary for this Article is available as a Supplementary Information file. Source data are provided with this paper.

## Code availability
Code to reproduce the analyses described in this manuscript can be accessed via: https://github.com/HobartJoe/Human_Keloid_scRNAseq. Code for the analysis of scRNA-seq data is also provided in a Zenodo repository with the identifier (https://doi.org/10.5281/zenodo.4784648)[61].

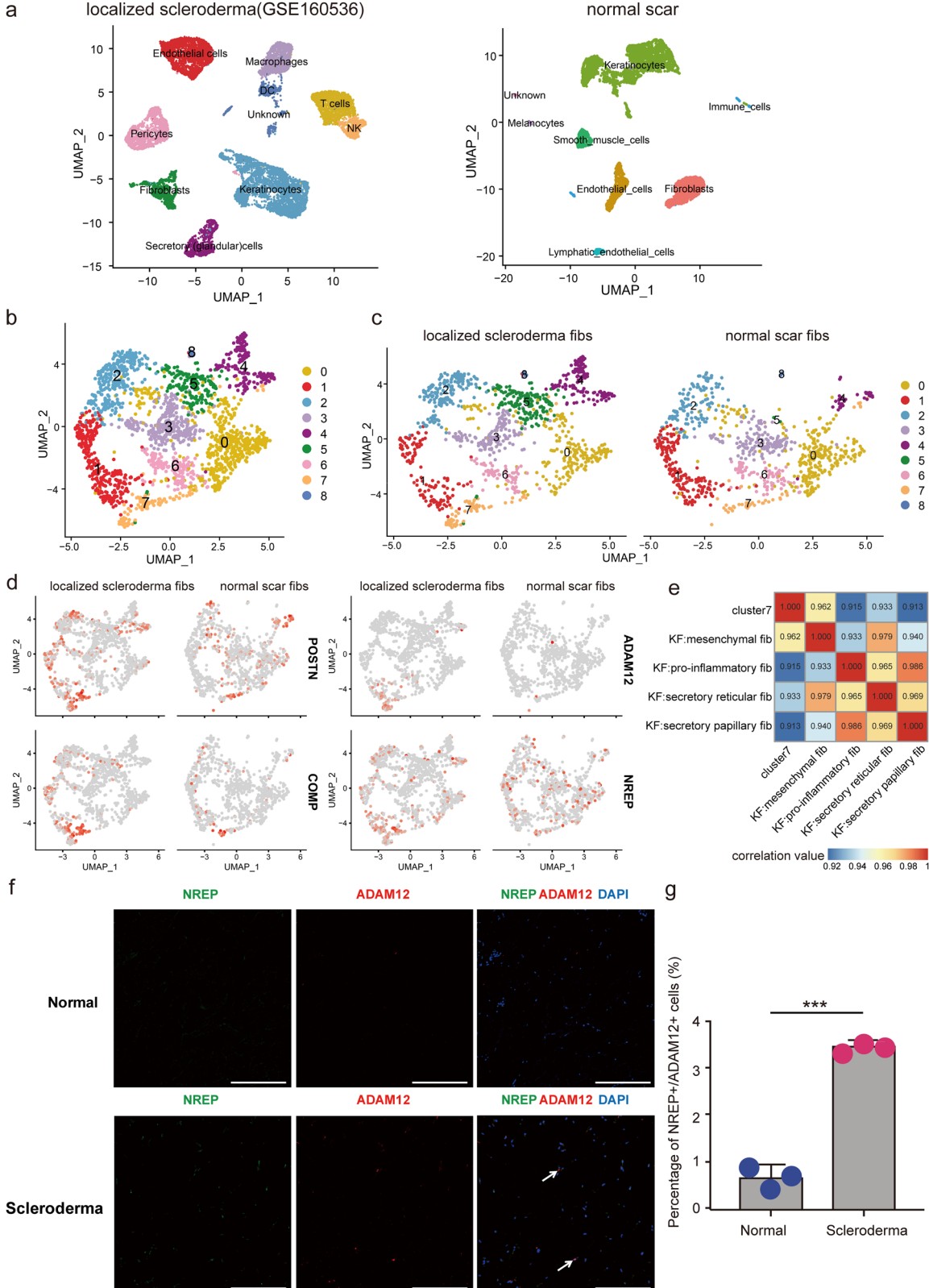

**Fig. 7 Mesenchymal fibroblasts are increased in scleroderma. a** UMAP visualization of dermal skin cell populations of patients with scleroderma (data from NCBI Gene Expression Omnibus (GSE160536)) in the left panel, and a normal scar (NS3) sample from our data is showed on the right. **b** Integration analysis of two sources of fibroblasts, one from scleroderma, and the other one from a normal scar. **c** Divided UMAP visualization of the integration results. **d** Feature plots of the expression distribution of mesenchymal signature genes in scleroderma and normal scar fibroblasts. **e** Correlation analysis of the average expression level of cluster 7 cells (scleroderma) and four types of fibroblasts from keloid samples. **f** Immunofluorescence staining of NREP and ADAM12 in normal and scleroderma tissues. Arrowheads indicate NREP+/ADAM12+ cells. Scale bar = 200 μm. **g** Percentage of NREP+/ADAM12+ cells in normal and scleroderma tissues. Data are presented as mean values ± SD ($n = 5$ images examined over 3 independent experiments). Two-sided unpaired $t$-test, ***$P = 0.00013$.

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

## Acknowledgements

This work was supported by grants from the National Natural Science Foundation of China (82073418, 81903189) and the Natural Science Foundation of Guangdong Province (2018A030310464).

## Author contributions

Conceptualization: B.Y., Z.R., and C.-C.D.; methodology: B.Y., Z.R., D.W., Y.-P.X., Y.-F.H., and C.-C.D.; formal analysis: C.-C.D., Y.-F.H., and D.-H.Z.; investigation: C.-C.D., Y.-F.H., D.-H.Z., Q.C., Q.-L.F., J.-J.G., and L.-X.Z.; writing-original draft preparation: Z.R., C.-C.D., and Y.-F.H.; writing-review and editing: C.-C.D., Z.-R., and B.Y.

## Competing interests

The authors declare no competing interests.
