## [Peer Review File · Nature Communications]

REVIEWER COMMENTS

Reviewer #1 (Remarks to the Author):

The manuscript by Deng, Hu, and Zhu et al. describe the cell types and cell-cell communication of keloid versus normal scar fibroblasts. The fibroblasts in both backgrounds largely fit into the four known fibroblast subpopulations in the dermis. They further show that mesenchymal fibroblasts are enriched in keloid scars and that they produce extra collagen I and III fibers. This positive trend of mesenchymal fibroblasts was also found in scleroderma. Overall, this work is well done and the timing is good considering multiple competing studies are in the works or posted to preprint servers. I have a few concerns with data analysis detailed below that should be addressed before publication.

1) The data needs to be deposited into a publicly available database. It is not sufficient for the data to be available from the corresponding author upon reasonable request.

2) The ratios of cell populations in Figure 1 are determined with keratinocytes in the mix, despite the epidermis purposely being removed before sequencing. Although it is very difficult to completely remove the adjacent tissue and some cells would be expected, the range of keratinocytes within each sample varies greatly. Keratinocytes should be removed from the proportion analysis and the ratios redone to limit artificial signal from unequal dissociation.

3) Figure 3 shows side-by-side comparisons of fibroblasts in normal scars and keloids. POSTN and COL11A1 are shown in feature plots to suggest they are expressed more in keloid mesenchymal fibroblasts than in normal scar fibroblasts. However, these cells are largely absent in normal scar fibroblasts which could artificially give the overexpression result. The authors should also show a graph that depicts the average cell expression of each gene in mesenchymal fibroblasts between the two conditions to verify that these two genes are in fact overexpressed specifically in keloids.

4) Myofibroblasts are a common fibroblast state during wound healing, however, the authors do not show them as one of the cell types. Are myofibroblasts present in either normal scar or keloids? Are they contributing ECM in the scars?

5) Several studies have analyzed fibroblasts in wound healing situations in the mouse. How does the human environment compare to what is found in mice, such as what is shown in Guerrero-Juarez et al. 2019 Nat Comm and Gay et al. 2020 Science Advances?

6) Minor grammatical errors are littered throughout the manuscript.

Reviewer #2 (Remarks to the Author):

This paper looks to molecularly differentiate keloid from normal scars based on sc-RNAseq analysis of fibroblast subpopulations.

The sample description by the authors is unacceptably opaque. Consider the opening sentences of the result section: "We only used the dermis for scRNA-seq analysis because keloid represents a skin dermis fibrotic disease. After stringent quality control (fig. S1A and B), we obtained transcriptomes of 40655 cells (keloid: 21488; normal scar: 19167)." Figure 1 seems to show that 3 keloid scars were compared to 3 "normal scars" but the Methods indicate 3 keloid donor and 2 normal scar donors. Were 2 of the normal scars obtained from a single donor, a problematic confounding effect in a study

already so underpowered? How were keloid and normal scars clinically or histopathologically distinguished? Is there any comment on the genetic background/gender of the patients?

Once the fibroblast clusters are identified ((C2, C4, C8, C14, C15), the authors on line 121 of page 5 switch to a new naming convention (FIB3?) without discussing in the text how the two naming systems map in the text.

I conclude in Figure 2 that the authors have regrouped all the cells from the 5 fibroblast clusters and clustered them into 13 new clusters. If so, why perform any analysis at the FIB3 level previously, if the actual analysis is happening on the 13 new clusters? What is the quantitative basis on which the hierarchical analysis in Figure 2D was used to come up with 4 groupings. Was it purely visual? How does Figure 2E: "demonstrate that sC2, sC3 and sC9 were pro- inflammatory fibroblasts, sC6 and sC7 were secretory-papillary fibroblasts, sC1 and sC4 were mesenchymal fibroblasts and sC5 were secretory-reticular fibroblasts (Fig. 2E and fig. S2A)."? If Figure S2A is the source of mapping these groupings to known categories, is there no quantitative metric but purely visual analysis used?

The remainder of the paper is dedicated to the following work:

- Keloids are significantly increased in so-called keloid mesenchymal fibroblasts
- Some GO and tx-factor network analysis of the differentially expressed genes in the relevant subpopulation
- An attempt to validate this increased mesenchymal population using two antibodies (CD266 and CD9) whose specificity between the subpopulations is not obviously pre-established (Figure 3C)
Off-the-shelf ligand/receptor analysis showing the possibility of fibrotic interactions increased in the keloid fibroblasts
- An attempt to flow sort the keloid mesenchymal fibroblasts based on CD266 and CD9 with validation by bulk RNAseq, I believe, not a repeat scRNAseq that could place the cells in context of the earlier experiments and provide a measure of specificity
- An experimental validation "We next collected the supernatant of CD266+/CD9 or other fibroblasts to treat other fibroblasts." What I believe the authors are claiming is that they used supernatant from the keloid mesenchymal fibroblasts to stimulate collagen synthesis in fibroblasts not harboring these markers. This is the most important part of the study, Figure 5H, in which the authors state that they inhibited collagen production with an POSTN antibody. But how many independent primary fibroblast donors were used? There is a Methods section detailing 'all experiments were performed in triplicate' and statistical methods, but none of the critical blocking experiments appear to have associated numbers in the text or in the Figure, only a single Western. How many total replicates were performed?
- A similar mesenchymal fibroblast population was identified in a prior single cell study of scleroderma.

- Minor points

The way "lineage expansion" is used to describe more endothelial cells in line 115 of page 5 is very misleading. The authors are not tracing lineages here. Just call it an increase in endothelial cells.

Reviewer #3 (Remarks to the Author):

This is a technical tour-de-force to identify and characterize fibroblastic subpopulations in fibrotic skin diseases, particularly keloids. The authors identify four principle subpopulations of fibroblastic cells by single-cell RNA-Seq. The results indicate that among the four types of subpopulations, i.e., secretory-papillary, secretory-reticular, mesenchymal and pro-inflammatory, the proportion of mesenchymal fibroblast subpopulation was significantly increased, providing a plausible explanation for tissue fibrosis in keloids.

There are a few suggestions for clarifications which would improve the quality of the manuscript.

1. Identification of functional subpopulation of cells with fibroblastic lineage is based on utilization of a number of markers. It is not quite clear how specific these expression markers are in discriminating the four different populations. For example, how specific is COL11A1 and POSTN expression for so-called mesenchymal progenitor cells? Similarly, how strong is the evidence that expression of COL13A1, COL18A1 and COL23A1 discriminate "papillary" fibroblasts from mesenchymal cells? The authors indicate (page 7) that these are "known markers" of papillary fibroblasts, but there is no reference to that statement.

2. While the investigators were able to demonstrate fibroblastic subpopulations with differential gene expression, the study does not take into account the morphologic heterogeneity at tissue level that is well established within keloids, some parts of the lesions being highly fibrotic with accumulation of tightly packed collagen fibers, while other areas can be seen to be occupied by dense inflammatory cell infiltrates. One would expect that the characteristics of isolated cell populations in these different areas are quite distinct. Consequently, it would be important to have some sort of histopathologic correlate of the areas of the lesions that were biopsied for isolation of cell suspensions. In this context, the study is somewhat preliminary in the sense that only three keloids were used for the study, and there is no morphologic correlations of the area subjected to biopsy. In fact, there is very little clinical description of the keloids studied, for example, were they mature, growing, recently developed? Was the sampling done from the center or from the edge of the lesion? It is stated that no patient received chemotherapy or radiotherapy prior to surgery, but it is not clear whether they were treated with intralesional steroids, the standard treatment for keloids.

3. While the keloids were probably isolated from Chinese patients, it would be important to know this, because the keloids in Asian patients are morphologically different from those encountered, for example, of those of African ancestry, and the genetic background may impact the cellular identity in these lesions.

4. For the scientists who have traditionally worked on fibrotic diseases, the distinction of mesenchymal and pro-inflammatory fibroblasts is somewhat confusing, as both these types were selected by using COL1A1 as a marker. Traditionally, the collagen over-producing cells in fibrotic diseases are traditionally described as myofibroblasts with α smooth muscle actin and vimentin as specific markers. Was expression of these markers checked, and if so, can the population of "mesenchymal" fibroblasts can be equated with myofibroblasts? Clarification of this nomenclature would allow the reader to put the findings of this study to prospective with the extensive literature that exist on cellular heterogeneity in keloids and other fibrotic diseases.

5. Considering the importance of this study towards understanding the pathomechanism of keloids and potentially other fibrotic skin diseases, a brief comment to their therapeutic perspective, either regarding the currently used treatment modalities, and perhaps some novel approaches, would be helpful to give the translational perspective to this study.

Reviewer #1 (Remarks to the Author):

The manuscript by Deng, Hu, and Zhu et al. describe the cell types and cell-cell communication of keloid versus normal scar fibroblasts. The fibroblasts in both backgrounds largely fit into the four known fibroblast subpopulations in the dermis. They further show that mesenchymal fibroblasts are enriched in keloid scars and that they produce extra collagen I and III fibers. This positive trend of mesenchymal fibroblasts was also found in scleroderma. Overall, this work is well done and the timing is good considering multiple competing studies are in the works or posted to preprint servers. I have a few concerns with data analysis detailed below that should be addressed before publication.

1) The data needs to be deposited into a publicly available database. It is not sufficient for the data to be available from the corresponding author upon reasonable request.

As suggested, we have deposited all RNA-Seq and scRNA-seq raw data into Gene Expression Omnibus with accession codes PRJNA688983 (for reviewer link: <https://dataview.ncbi.nlm.nih.gov/object/PRJNA688983?reviewer=gkuaeqnogecaaar1hknb3o1qk7>) and GSE163973 (the reviewer token: qzgfkcscrtwdrgp), respectively. All the data will be publicly available after Dec. 2021. The raw data underlying Figs. 3g, 5b, c, f, g, h and 7g (including qRT-PCR, Western Blot, Immunofluorescence data) have been provided as a “Source Data”, please see the “Source Data” file.

2) The ratios of cell populations in Figure 1 are determined with keratinocytes in the mix, despite the epidermis purposely being removed before sequencing. Although it is very difficult to completely remove the adjacent tissue and some cells would be expected, the range of keratinocytes within each sample varies greatly. Keratinocytes should be removed from the proportion analysis and the ratios redone to limit artificial signal from unequal dissociation.

We thank the reviewer for the valuable suggestion. As suggested, we have removed the cells in epidermis (keratinocytes and melanocytes) and redone the proportion and ratio analyses. Removing epidermis cells doesn't change the major conclusion that the proportions of endothelial cells and smooth muscle cells are increased and the proportions of fibroblasts are

decreased in keloid tissues compared to normal scar tissues. Please see Figure 1f.

3) Figure 3 shows side-by-side comparisons of fibroblasts in normal scars and keloids. POSTN and COL11A1 are shown in feature plots to suggest they are expressed more in keloid mesenchymal fibroblasts than in normal scar fibroblasts. However, these cells are largely absent in normal scar fibroblasts which could artificially give the overexpression result. The authors should also show a graph that depicts the average cell expression of each gene in mesenchymal fibroblasts between the two conditions to verify that these two genes are in fact overexpressed specifically in keloids.

The reviewer raises a good point. As suggested, we have shown a graph that depicts the average cell expression of POSTN and COL11A1 in mesenchymal fibroblasts between keloids and normal scars. The results suggest that not only the numbers of mesenchymal fibroblasts, but also the average cell expressions of POSTN are higher in keloids than in normal scars ($P=0.011$). The average cell expressions of COL11A1 are also higher in keloids than in normal scars although the difference is not significant ($P=0.16$). Please see Figure 3a.

4) Myofibroblasts are a common fibroblast state during wound healing, however, the authors do not show them as one of the cell types. Are myofibroblasts present in either normal scar or keloids? Are they contributing ECM in the scars?

We appreciate the reviewer for raising a very interesting point. The number of myofibroblasts have been reported to be increased at the early phase of wound healing, and decreased at the late phase of wound healing when scar forms (Bochaton-Piallat et al., 2016; Hinz, 2016). Although Ehrlich et al. (1994) suggested that the absence of myofibroblasts as a feature that differentiates keloids from hypertrophic scars (Ehrlich et al., 1994), the opposite has also been observed (Santucci et al., 2001). Overwhelming majority of studies report the presence of α -SMA positive myofibroblasts in 33–81% of the keloids analyzed (Amadeu et al., 2003; Kamath et al., 2002; Lee et al., 2004; Limandjaja et al., 2020; Santucci et al., 2001). To check whether myofibroblasts are present in normal scar or keloids, we used ACTA2 gene (encoding α -SMA), a marker of myofibroblasts (Griffin et al., 2020), to identify myofibroblasts in our scRNA-seq data. We found that the number of myofibroblasts

increased in keloids compared to normal scars ($26.0\% \pm 4.3\%$ vs $13.3\% \pm 6.4\%$) (Supplementary Fig. 4a). In keloids, myofibroblasts were enriched in mesenchymal fibroblast subpopulation ($53.8\% \pm 9.2\%$) and also existed in other three fibroblast subpopulations (pro-inflammatory fibroblast: $29.7\% \pm 10.6\%$; secretory-papillary fibroblast: $8.9\% \pm 1.2\%$; secretory-reticular fibroblast: $7.6\% \pm 0.3\%$) (Supplementary Fig. 4a). We also analyzed the expression of α -SMA and ADAM12 (a marker of mesenchymal fibroblast) in keloid and normal scar tissues by immunofluorescence. The immunofluorescence experiments showed similar results as single cell sequencing results. We detected α -SMA positive vascular smooth muscle cells (α -SMA is also the marker of vascular smooth muscle cells (Griffin et al., 2020)) and hardly detected α -SMA positive myofibroblasts in normal scar tissues (Supplementary Fig. 4b). In keloid we can detect both α -SMA positive smooth muscle cells and myofibroblasts (Supplementary Fig. 4b).

Myofibroblasts have been proven to be the major contributor of ECM in the early phase of wound healing and the number of myofibroblast decreases when wound healing progress into the remodeling stage in mice (Bochaton-Piallat et al., 2016; Hinz, 2016). In human abnormal mature scar, such as keloid, the role of myofibroblasts in ECM deposition is not very clear because of lack of the animal models (Andrews et al., 2016; Griffin et al., 2020). Collagen I, III and V are major collagen types in keloid ECM formation (Macarak et al., 2021). Our collagen expression pattern analyses indicate that the mesenchymal fibroblasts and secretory-reticular fibroblasts are major contributors for collagen I, III and V expression in keloid (Supplementary Fig. 2f). We find that 61.4% of myofibroblasts are in the mesenchymal fibroblast and secretory-reticular fibroblast subpopulations. These myofibroblasts may contribute ECM in keloid as mesenchymal and secretory-reticular fibroblasts do. Therefore, our discovery is consistent with previous hypothesis that myofibroblasts contribute to ECM formation in the scars but the difference is that our discovery might expand the myofibroblast hypothesis. Our discovery indicates that mesenchymal fibroblasts and secretory-reticular fibroblasts are contributors of ECM deposition in keloid, and myofibroblasts (as a subpopulation of mesenchymal and secretory-reticular fibroblasts) also contribute to ECM formation. It is worth to note that this discrepancy may also reflect the difference between tissues in the early phase of wound

healing in the mice researches and mature scars of our human research or reflect species difference between human and mice. We have added these results to our manuscript. Please see Supplementary Fig. 4 and page 10 and 19.

References

- Amadeu, T., Braune, A., Mandarim-de-Lacerda, C., Porto, L.C., Desmoulière, A., and Costa, A. (2003). Vascularization pattern in hypertrophic scars and keloids: a stereological analysis. *Pathology, research and practice* 199, 469-473.
- Andrews, J.P., Marttala, J., Macarak, E., Rosenbloom, J., and Uitto, J. (2016). Keloids: The paradigm of skin fibrosis - Pathomechanisms and treatment. *Matrix biology : journal of the International Society for Matrix Biology* 51, 37-46.
- Bochaton-Piallat, M.L., Gabbiani, G., and Hinz, B. (2016). The myofibroblast in wound healing and fibrosis: answered and unanswered questions. *F1000Research* 5.
- Ehrlich, H.P., Desmoulière, A., Diegelmann, R.F., Cohen, I.K., Compton, C.C., Garner, W.L., Kapanci, Y., and Gabbiani, G. (1994). Morphological and immunochemical differences between keloid and hypertrophic scar. *The American journal of pathology* 145, 105-113.
- Griffin, M.F., desJardins-Park, H.E., Mascharak, S., Borrelli, M.R., and Longaker, M.T. (2020). Understanding the impact of fibroblast heterogeneity on skin fibrosis. *Disease models & mechanisms* 13.
- Hinz, B. (2016). The role of myofibroblasts in wound healing. *Current research in translational medicine* 64, 171-177.
- Kamath, N.V., Ormsby, A., Bergfeld, W.F., and House, N.S. (2002). A light microscopic and immunohistochemical evaluation of scars. *Journal of cutaneous pathology* 29, 27-32.
- Lee, J.Y., Yang, C.C., Chao, S.C., and Wong, T.W. (2004). Histopathological differential diagnosis of keloid and hypertrophic scar. *The American Journal of dermatopathology* 26, 379-384.
- Limandjaja, G.C., Niessen, F.B., Scheper, R.J., and Gibbs, S. (2020). The Keloid Disorder: Heterogeneity, Histopathology, Mechanisms and Models. *Frontiers in cell and developmental biology* 8, 360.
- Macarak, E.J., Wermuth, P.J., Rosenbloom, J., and Uitto, J. (2021). Keloid disorder: Fibroblast differentiation and gene expression profile in fibrotic skin diseases. *Experimental dermatology* 30, 132-145.
- Santucci, M., Borgognoni, L., Reali, U.M., and Gabbiani, G. (2001). Keloids and hypertrophic scars of Caucasians show distinctive morphologic and immunophenotypic profiles. *Virchows Archiv : an*

5) Several studies have analyzed fibroblasts in wound healing situations in the mouse. How does the human environment compare to what is found in mice, such as what is shown in Guerrero-Juarez et al. 2019 Nat Comm and Gay et al. 2020 Science Advances?

We thank the reviewer for providing the good suggestion. We compared our data with the data in mice from Guerrero-Juarez et al. 2019 Nat Comm and Gay et al. 2020 Science Advances. We performed unsupervised clustering on all wound fibroblasts from the Nat Comm article and observed heterogeneity with 12 subclusters, sC1 through sC12 (Fig. R1a), which was consistent with the authors' original findings. We next analyzed the expression of 4 fibroblast subpopulation markers from our research (Supplementary Fig. 2e) in the wound fibroblast subclusters. We found that most of the wound fibroblasts expressed the mesenchymal fibroblast marker Postn, Aspn and reticular fibroblast marker Mfap5 (Fig. R1b). We can nearly not detect the expression of the mesenchymal fibroblast marker Comp, papillary fibroblast marker Apcdd1, Col13a1 and pro-inflammatory fibroblast marker Ccl19 in the wound fibroblasts (Fig. R1b). We also performed unsupervised clustering on all fibrotic hairless scar fibroblasts from the Science Advances article (Fig. R1c). Similar to the Nat Comm article, most of the fibrotic hairless scar fibroblasts expressed the markers Postn, Aspn and Mfap5, and did not express the markers Comp, Apcdd1, Col13a1 and Ccl19 (Fig. R1d). These results suggest that in the articles of Guerrero-Juarez et al. 2019 Nat Comm and Gay et al. 2020 Science Advances, the fibroblasts in wound healing situations in the mouse can't be subclustered into 4 subpopulations by the markers of our paper. This discrepancy is likely due to the difference between fresh scars of the wound in the mice researches and mature scars of our human research. It may also be due to species difference between humans and mice.

Figure R1. Analyzing the data in mice from Guerrero-Juarez et al. 2019 Nat Comm and Gay et al. 2020 Science Advances with the fibroblast subpopulation markers we identified. (a) Subclustering of wound fibroblasts from the Nat Comm article identified 12 distinct subtypes. Color-coded UMAP plot is shown and each fibroblast subcluster (sC1 through sC12) is defined on the right. **(b)** Feature plots of representative marker genes of each fibroblast subpopulation in (a). Expression levels for each cell are color-coded and overlaid onto UMAP plot. **(c)** Subclustering of fibrotic hairless scar fibroblasts from the Science Advances article identified 13 distinct subtypes. **(d)** Feature plots of representative marker genes of each fibroblast subpopulation in (c). Expression levels for each cell are color-coded and overlaid onto UMAP plot.

6) Minor grammatical errors are littered throughout the manuscript.

We apologize for the errors. We have carefully proofread the manuscript and changed these errors, besides it has been edited by an English-editing company (American Journal Experts). The editorial certificate is shown below.

Reviewer #2 (Remarks to the Author):

This paper looks to molecularly differentiate keloid from normal scars based on sc-RNAseq analysis of fibroblast subpopulations.

The sample description by the authors is unacceptably opaque. Consider the opening sentences of the result section: “We only used the dermis for scRNA-seq analysis because keloid represents a skin dermis fibrotic disease. After stringent quality control (fig. S1A and B), we obtained transcriptomes of 40655 cells (keloid: 21488; normal scar: 19167).” Figure 1 seems to show that 3 keloid scars were compared to 3 “normal scars” but the Methods indicate 3 keloid donor and 2 normal scar donors. Were 2 of the normal scars obtained from a single donor, a problematic confounding effect in a study already so underpowered? How were keloid and normal scars clinically or histopathologically distinguished? Is there any comment on the genetic background/gender of the patients?

We are very sorry for the mistake. In our first draft of our paper, we collected 2 normal scar tissues from 2 different donors, and we added a normal scar tissue from another donor when we finished the second version of our draft. Therefore, we collected 3 normal scar tissues from 3 different donors at last, but in the manuscript, we forget to change the donor number

from 2 to 3. We correct this mistake and add more sample information, including the information about how we distinguished keloid and normal scars and the genetic background/gender of the patients. Please see the MATERIALS AND METHODS (page 21-22) and Supplementary Table 1.

Once the fibroblast clusters are identified ((C2, C4, C8, C14, C15), the authors on line 121 of page 5 switch to a new naming convention (FIB3?) without discussing in the text how the two naming systems map in the text.

We are sorry for making the confusion. To make it clear, we no longer use the new naming system in Figure 1g. Instead, we use the cell type names consistent with figure 1d-f. Please see figure 1g.

I conclude in Figure 2 that the authors have regrouped all the cells from the 5 fibroblast clusters and clustered them into 13 new clusters. If so, why perform any analysis at the FIB3 level previously, if the actual analysis is happening on the 13 new clusters? What is the quantitative basis on which the hierarchical analysis in Figure 2D was used to come up with 4 groupings. Was it purely visual? How does Figure 2E: “demonstrate that sC2, sC3 and sC9 were pro-inflammatory fibroblasts, sC6 and sC7 were secretory-papillary fibroblasts, sC1 and sC4 were mesenchymal fibroblasts and sC5 were secretory-reticular fibroblasts (Fig. 2E and fig. S2A).”? If Figure S2A is the source of mapping these groupings to known categories, is there no quantitative metric but purely visual analysis used?

We are sorry for making the confusion in Figure 1g. As mentioned in the previous response, we no longer use the new naming system in Figure 1g and use the cell type names consistent with figure 1d-f. When we explore the number of differentially expressed genes between keloid and normal scar clusters, we find that fibroblast has the largest difference, suggesting that fibroblasts undergo significant changes during the fibrotic progress. This result prompts us to focus on fibroblasts in our next work and thus classify fibroblasts into more subclusters to explore their functions in detail.

We classified the fibroblasts into 4 subpopulations, which was based on the hierarchical cluster analysis in Figure 2d and Supplementary Fig. 2a-d. We used quantitative analyses to

come up with 4 subpopulations in Figure 2d. The average gene expression value of all clusters in fibroblasts were input to conduct agglomerative hierarchical clustering with a Euclidean distance metric and Ward's minimum variance algorithm (Babbin et al., 2015; Grisanzio et al., 2018). The optimal number of clusters was determined using four standard methods including the silhouette coefficient (Rousseeuw, 1987), Calinski–Harabasz index (Wang and Xu, 2019), Davies–Bouldin index (Davies and Bouldin, 1979), and dendrogram (Van Soest et al., 2012). We identified the four-cluster was the optimal number of clusters (shown in Supplementary Fig. 2a-d), and this is consistent with the expression of specific markers from a published article (Sole-Boldo et al., 2020). Additionally, collagen producing is one of the major functions of fibroblasts, and the four-population clusterization is also consistent with collagen expression pattern (Supplementary Fig. 2f). We have modified the figures and manuscript. Please see Fig. 2d, Supplementary Fig. 2 and page 6-7.

References

- Babbin, S.F., Velicer, W.F., Paiva, A.L., Brick, L.A., and Redding, C.A. (2015). Replicating cluster subtypes for the prevention of adolescent smoking and alcohol use. *Addictive behaviors* 40, 57-65.
- Davies, D.L., and Bouldin, D.W. (1979). A cluster separation measure. *IEEE transactions on pattern analysis and machine intelligence* 1, 224-227.
- Grisanzio, K.A., Goldstein-Piekarski, A.N., Wang, M.Y., Rashed Ahmed, A.P., Samara, Z., and Williams, L.M. (2018). Transdiagnostic Symptom Clusters and Associations With Brain, Behavior, and Daily Function in Mood, Anxiety, and Trauma Disorders. *JAMA psychiatry* 75, 201-209.
- Rousseeuw P. J. (1987). Silhouettes: A graphical aid to the interpretation and validation of cluster analysis. *J. Comput. Appl. Math.* 20 53–65. [10.1016/0377-0427\(87\)90125-7](https://doi.org/10.1016/0377-0427(87)90125-7)
- Sole-Boldo, L., Raddatz, G., Schutz, S., Mallm, J.P., Rippe, K., Lonsdorf, A.S., Rodriguez-Paredes, M., and Lyko, F. (2020). Single-cell transcriptomes of the human skin reveal age-related loss of fibroblast priming. *Commun Biol* 3, 188.
- Van Soest, R.W., Boury-Esnault, N., Vacelet, J., Dohrmann, M., Erpenbeck, D., De Voogd, N.J., Santodomingo, N., Vanhoorne, B., Kelly, M., and Hooper, J.N. (2012). Global diversity of sponges (Porifera). *PloS one* 7, e35105.
- Wang X., Xu Y. (2019). An improved index for clustering validation based on Silhouette index and

The remainder of the paper is dedicated to the following work:

- Keloids are significantly increased in so-called keloid mesenchymal fibroblasts
- Some GO and tx-factor network analysis of the differentially expressed genes in the relevant subpopulation
- An attempt to validate this increased mesenchymal population using two antibodies (CD266 and CD9) whose specificity between the subpopulations is not obviously pre-established (Figure 3C)

Thanks for the questions. In Figure 3c, we analyzed the expression of CD266 (encoded by *TNFRSF12A* gene) and CD9 in four fibroblast populations, we found that CD266 was high-expressed and CD9 was low-expressed in mesenchymal fibroblast subpopulation compared to other fibroblast populations, so we chose these two markers for flow cytometry sorting. We also considered other membrane markers, such as ADAM12 and SDC1, but we couldn't find good commercially available flow cytometry antibodies for these markers. Considering expression specificity and antibody availability, we chose CD266 and CD9 for further flow cytometry sorting at last. qRT-PCR, Western blot and RNA-seq validated that mesenchymal fibroblasts marker genes and mesenchymal associated functions were significantly enriched in CD266+/CD9- fibroblasts compared to other fibroblasts (Fig. 5, b-e), suggesting that CD266 and CD9 antibodies were effective to isolate mesenchymal fibroblasts.

Off-the-shelf ligand/receptor analysis showing the possibility of fibrotic interactions increased in the keloid fibroblasts

- An attempt to flow sort the keloid mesenchymal fibroblasts based on CD266 and CD9 with validation by bulk RNAseq, I believe, not a repeat scRNAseq that could place the cells in context of the earlier experiments and provide a measure of specificity

We appreciate the reviewer for the comments. qRT-PCR, Western blot and RNA-seq validated that mesenchymal fibroblast marker genes and mesenchymal associated functions were significantly enriched in CD266+/CD9- fibroblasts compared to other fibroblasts (Fig. 5, b-e), suggesting that CD266 and CD9 antibodies were effective to isolate mesenchymal fibroblasts. To further validate the specificity of CD266 and CD9, we calculate the similarity

of keloid mesenchymal fibroblasts transcriptome measured by bulk RNAseq and scRNA-seq (Cao et al., 2019; Crowell et al., 2020; Hay et al., 2018). In brief, we first calculated the average expression level of mesenchymal fibroblast subpopulation (MF) and other three fibroblast subpopulations (other fibroblasts) in each keloid sample based on scRNAseq data. Then, normalized the gene expression measurements for two bulk RNAseq data by the total expression, multiplied this by a scale factor (10000) (Butler et al., 2018). Next, we used Combat method (from R package sva) (Leek et al., 2012) to correct the batch effect in the combined dataset. After removing known batch effects, we performed the principal component analysis (PCA) and calculated the Pearson correlation coefficients between two sources of expression data. The results showed that the gene expression of CD266+/CD9- fibroblasts were most like the gene expression of mesenchymal fibroblast subpopulation (MF), and the gene expression of other fibroblasts here were most like other three fibroblast subpopulations (other fibroblasts) (Fig. R2). Taken together, we believe that the analyses in Fig. 5b-e and Fig. R2 can demonstrate the efficacy of using CD266+/CD9- to isolate the keloid mesenchymal fibroblasts.

Figure R2. Validation reliability of captured fibroblasts by the CD266+/CD9- markers. (a) Principal Component Analysis of the merged two groups of fibroblasts in scRNA-seq data and two types of fibroblasts in bulk RNA-seq data. Open circles represent data from bulk RNA-seq, and colored shapes represent data from single-cell data, two colors represent two group fibroblasts. MF: mesenchymal fibroblasts, other fibroblasts: the rest of fibroblasts in keloid excluding mesenchymal fibroblasts. **(b)** Correlation analysis of the expression profile of two sources of fibroblasts, including scRNA-seq and bulk RNA-seq.

References

Butler, A., Hoffman, P., Smibert, P., Papalexi, E., and Satija, R. (2018). Integrating single-cell transcriptomic data across different conditions, technologies, and species. *Nature biotechnology* 36, 411-420.

Cao, J., Spielmann, M., Qiu, X., Huang, X., Ibrahim, D.M., Hill, A.J., Zhang, F., Mundlos, S., Christiansen, L., Steemers, F.J., *et al.* (2019). The single-cell transcriptional landscape of mammalian organogenesis. *Nature* 566, 496-502.

Crowell, H.L., Soneson, C., Germain, P.L., Calini, D., Collin, L., Raposo, C., Malhotra, D., and Robinson, M.D. (2020). muscat detects subpopulation-specific state transitions from multi-sample multi-condition single-cell transcriptomics data. *Nature communications* 11, 6077.

Hay, S.B., Ferchen, K., Chetal, K., Grimes, H.L., and Salomonis, N. (2018). The Human Cell Atlas bone marrow single-cell interactive web portal. *Experimental hematology* 68, 51-61.

Leek, J.T., Johnson, W.E., Parker, H.S., Jaffe, A.E., and Storey, J.D. (2012). The sva package for removing batch effects and other unwanted variation in high-throughput experiments. *Bioinformatics (Oxford, England)* 28, 882-883.

- An experimental validation “We next collected the supernatant of CD266+/CD9 or other fibroblasts to treat other fibroblasts.” What I believe the authors are claiming is that they used supernatant from the keloid mesenchymal fibroblasts to stimulate collagen synthesis in fibroblasts not harboring these markers. This is the most important part of the study, Figure 5H, in which the authors state that they inhibited collagen production with an POSTN antibody. But how many independent primary fibroblast donors were used? There is a Methods section detailing ‘all experiments were performed in triplicate’ and statistical methods, but none of the critical blocking experiments appear to have associated numbers in the text or in the Figure, only a single Western. How many total replicates were performed?

We thank the reviewer for the comment. We repeated the experiments in Figure 5h three times with three different fibroblast donors, and the results were consistent. The other two results were shown in Figure R3. We have added the replication number of the experiments to the text and Figure legends, please see page 30 and 46.

Figure R3. POSTN neutralizing antibody inhibited the increased expression of collagen I and III in other fibroblasts upon treatment of CD266+/CD9- supernatant. Keloid other fibroblasts were treated as indicated in the figure. The expression of collagen I and collagen III were analyzed by Western Blot. (a) and (b) showed the results from two different primary fibroblast donors.

- A similar mesenchymal fibroblast population was identified in a prior single cell study of scleroderma.

- Minor points

The way “lineage expansion” is used to describe more endothelial cells in line 115 of page 5 is very misleading. The authors are not tracing lineages here. Just call it an increase in endothelial cells.

According to the suggestion, we have changed “Lineage expansion” to “Increased proportions”. Please see page 5.

Reviewer #3 (Remarks to the Author):

This is a technical tour-de-force to identify and characterize fibroblastic subpopulations in fibrotic skin diseases, particularly keloids. The authors identify four principle subpopulations of fibroblastic cells by single-cell RNA-Seq. The results indicate that among the four types of subpopulations, i.e., secretory-papillary, secretory-reticular, mesenchymal and pro-inflammatory, the proportion of mesenchymal fibroblast subpopulation was significantly increased, providing a plausible explanation for tissue fibrosis in keloids.

There are a few suggestions for clarifications which would improve the quality of the manuscript.

1. Identification of functional subpopulation of cells with fibroblastic lineage is based on utilization of a number of markers. It is not quite clear how specific these expression markers are in discriminating the four different populations. For example, how specific is COL11A1 and POSTN expression for so-called mesenchymal progenitor cells? Similarly, how strong is the evidence that expression of COL13A1, COL18A1 and COL23A1 discriminate “papillary” fibroblasts from mesenchymal cells? The authors indicate (page 7) that these are “known markers” of papillary fibroblasts, but there is no reference to that statement.

We thank the reviewer for this comment. We classify the fibroblasts into 4 subpopulations, which is based on the hierarchical cluster analysis in Figure 2d and Supplementary Fig. 2a-d. We find that the optimal clusterization is four-subpopulation (Supplementary Fig. 2a-d), which is consistent with a published article (Sole-Boldo et al., 2020) (Supplementary Fig. 2e). Additionally, collagen producing is one of the major functions of fibroblasts, and the four-population clusterization is also consistent with collagen expression pattern (Supplementary Fig. 2f). All of the markers used to discriminate the four different populations were reported by literatures and validated by the article (Sole-Boldo et al., 2020). POSTN and COL11A1 are associated with cartilage and bone development (Bonnet et al., 2016; Li et al., 2018; Sole-Boldo et al., 2020), respectively, were specifically expressed in the mesenchymal fibroblast subpopulation (Fig. 3a and Supplementary Fig. 2), suggesting a strong mesenchymal component for this cell subpopulation. COL13A1 (Peltonen et al., 1999; Sole-Boldo et al., 2020), COL18A1 (Philippeos et al., 2018; Sole-Boldo et al., 2020), and COL23A1 (Philippeos et al., 2018; Sole-Boldo et al., 2020) were reported as papillary fibroblast markers and specifically expressed in the secretory papillary fibroblast subpopulation (Supplementary Fig. 2). CCL19 (Förster et al., 2008; Sole-Boldo et al., 2020) and CXCL3 (Korbecki et al., 2021; Sole-Boldo et al., 2020) were reported as inflammatory cytokines and specifically expressed in the Pro-inflammatory fibroblast subpopulation (Supplementary Fig. 2e). Low collagen expression was also one of the characteristics of the Pro-inflammatory fibroblast subpopulation (Supplementary Fig. 2f) (Sole-Boldo et al., 2020). MFAP5 (Philippeos et al., 2018; Sole-Boldo et al., 2020) and WISP2 (Sole-Boldo et al., 2020)

were reported as reticular fibroblast markers and specifically expressed in the secretory reticular fibroblast subpopulation (Supplementary Fig. 2e). We have cited the associated literatures in the manuscript. Please see page 6-7 and the References.

References

Bonnet, N., Garnero, P., and Ferrari, S. (2016). Periostin action in bone. *Molecular and cellular endocrinology* 432, 75-82.

Förster, R., Davalos-Miszlitz, A.C., and Rot, A. (2008). CCR7 and its ligands: balancing immunity and tolerance. *Nature reviews Immunology* 8, 362-371.

Korbecki, J., Kojder, K., Kapczuk, P., Kupnicka, P., Gawrońska-Szklarz, B., Gutowska, I., Chlubek, D., and Baranowska-Bosiacka, I. (2021). The Effect of Hypoxia on the Expression of CXC Chemokines and CXC Chemokine Receptors-A Review of Literature. *International journal of molecular sciences* 22.

Li, A., Wei, Y., Hung, C., and Vunjak-Novakovic, G. (2018). Chondrogenic properties of collagen type XI, a component of cartilage extracellular matrix. *Biomaterials* 173, 47-57.

Peltonen, S., Hentula, M., Hägg, P., Ylä-Outinen, H., Tuukkanen, J., Lakkakorpi, J., Rehn, M., Pihlajaniemi, T., and Peltonen, J. (1999). A novel component of epidermal cell-matrix and cell-cell contacts: transmembrane protein type XIII collagen. *The Journal of investigative dermatology* 113, 635-642.

Philippeos, C., Telerman, S.B., Oulès, B., Pisco, A.O., Shaw, T.J., Elgueta, R., Lombardi, G., Driskell, R.R., Soldin, M., Lynch, M.D., et al. (2018). Spatial and Single-Cell Transcriptional Profiling Identifies Functionally Distinct Human Dermal Fibroblast Subpopulations. *The Journal of investigative dermatology* 138, 811-825.

Sole-Boldo, L., Raddatz, G., Schutz, S., Mallm, J.P., Rippe, K., Lonsdorf, A.S., Rodriguez-Paredes, M., and Lyko, F. (2020). Single-cell transcriptomes of the human skin reveal age-related loss of fibroblast priming. *Commun Biol* 3, 188.

2. While the investigators were able to demonstrate fibroblastic subpopulations with differential gene expression, the study does not take into account the morphologic heterogeneity at tissue level that is well established within keloids, some parts of the lesions being highly fibrotic with accumulation of tightly packed collagen fibers, while other areas can be seen to be occupied by dense inflammatory cell infiltrates. One would expect that the characteristics of isolated cell

populations in these different areas are quite distinct. Consequently, it would be important to have some sort of histopathologic correlate of the areas of the lesions that were biopsied for isolation of cell suspensions. In this context, the study is somewhat preliminary in the sense that only three keloids were used for the study, and there is no morphologic correlations of the area subjected to biopsy. In fact, there is very little clinical description of the keloids studied, for example, were they mature, growing, recently developed? Was the sampling done from the center or from the edge of the lesion? It is stated that no patient received chemotherapy or radiotherapy prior to surgery, but it is not clear whether they were treated with intralesional steroids, the standard treatment for keloids.

Thank the reviewer for the suggestions. All the keloids we used in this study were mature. Because the keloid samples in Chinese people are not very big, and we need a lot of cells for flow cytometry and 10×genomics single cell sequencing, we used all contents of the keloid samples, including the center and the edge of the samples, and mix them for further analysis. The patients were not treated with intralesional steroids before keloid excision, but after excision they were treated with intralesional steroids in our hospital. We have added more sample information, including the information about whether the keloids were mature, the location information the samples done from, and whether the keloid patients were treated with intralesional steroids. Please see the MATERIALS AND METHODS “Sample preparation and tissue dissociation” part (Please see page 21-22) and Supplementary Table 1.

3. While the keloids were probably isolated from Chinese patients, it would be important to know this, because the keloids in Asian patients are morphologically different from those encountered, for example, of those of African ancestry, and the genetic background may impact the cellular identity in these lesions.

Thanks for your advice. As you said, it’s important to know that the keloids in Asian patients are different from those in patients from another regions, such as in African ancestry patients. We have added the genetic background information of the patients in our study, please see the manuscript (page 21-22) and Supplementary Table 1.

4. For the scientists who have traditionally worked on fibrotic diseases, the distinction of mesenchymal and pro-inflammatory fibroblasts is somewhat confusing, as both these types were selected by using COL1A1 as a marker. Traditionally, the collagen over-producing cells in fibrotic diseases are traditionally described as myofibroblasts with α smooth muscle actin and vimentin as specific markers. Was expression of these markers checked, and if so, can the population of “mesenchymal” fibroblasts can be equated with myofibroblasts? Clarification of this nomenclature would allow the reader to put the findings of this study to prospective with the extensive literature that exist on cellular heterogeneity in keloids and other fibrotic diseases.

The reviewer raises an important point. As suggested, we analyzed the expression of ACTA2 (encoding α -SMA) and VIM (encoding VIMENTIN) in our single cell data. We found that in keloids, ACTA2 positive cells were enriched in mesenchymal fibroblast subpopulation (53.8% \pm 9.2%), and also existed in other three fibroblast subpopulations (pro-inflammatory fibroblast: 29.7% \pm 10.6%; secretory-papillary fibroblast: 8.9% \pm 1.2%; secretory-reticular fibroblast: 7.6% \pm 0.3%) (Supplementary Fig. 4a). Only part of mesenchymal fibroblasts was positive for ACTA2 expression (36.6% \pm 8.0%) (Supplementary Fig. 4a). All of fibroblasts expressed VIM (Figure R4), suggesting that VIM may be not a suitable marker for myofibroblasts. We also analyzed the expression of ADAM12 and α -SMA in keloid and normal scar tissues by immunofluorescence. The immunofluorescence experiments showed similar results as single cell sequencing results. In keloid we can detect both α -SMA positive smooth muscle cells and myofibroblasts, and only part of ADAM12 positive mesenchymal fibroblasts was α -SMA positive myofibroblasts (Supplementary Fig. 4b). These results suggested that a part of mesenchymal fibroblasts were myofibroblasts (36.6% \pm 8.0%), and most of myofibroblasts were in mesenchymal fibroblast subpopulation (53.8% \pm 9.2%). We have added these results to our manuscript. Please see Supplementary Fig. 4 and page 10 and 19. Please also refer to the response to question 4 raised by reviewer 1.

Figure R4. Feature plots of expression distribution for VIM in keloid fibroblasts (KL) and normal scar fibroblasts (NS). Expression levels for each cell are color-coded and overlaid onto the UMAP plot.

5. Considering the importance of this study towards understanding the pathomechanism of keloids and potentially other fibrotic skin diseases, a brief comment to their therapeutic perspective, either regarding the currently used treatment modalities, and perhaps some novel approaches, would be helpful to give the translational perspective to this study.

We thank the reviewer for the valuable suggestion. In most cases, keloid patients are effectively treated with non-targeted therapies such as surgical excision, but with varying degrees of recurrence. Our studies indicate that mesenchymal fibroblasts are important for the overexpression of collagens in keloid through POSTN. We may develop some methods, such as using small molecule inhibitors of POSTN, to target mesenchymal fibroblasts. Inhibiting or eliminating mesenchymal fibroblasts before or after non-targeted therapies in keloid patients may improve the therapeutic effect of keloid significantly. We have commented the therapeutic perspective of our study in Discussion part. Please see Page 20-21.

REVIEWERS' COMMENTS

Reviewer #1 (Remarks to the Author):

The authors have satisfied my critiques and have now presented a stronger manuscript that incorporates and enhances established literature on fibroblasts. This will be a helpful addition to the literature.

Reviewer #2 (Remarks to the Author):

The authors have clarified my primary concerns, especially the number of independent experiments used to ascertain the supernatant. I have no further concerns regarding the acceptance of this paper.

Reviewer #3 (Remarks to the Author):

The authors have appropriately revised the manuscript, and I recommend the manuscript for publication.

Reviewer #1 (Remarks to the Author):

The authors have satisfied my critiques and have now presented a stronger manuscript that incorporates and enhances established literature on fibroblasts. This will be a helpful addition to the literature.

Thank you very much for your nice comments.

Reviewer #2 (Remarks to the Author):

The authors have clarified my primary concerns, especially the number of independent experiments used to ascertain the supernatant. I have no further concerns regarding the acceptance of this paper.

Thank you very much for your nice comments.

Reviewer #3 (Remarks to the Author):

The authors have appropriately revised the manuscript, and I recommend the manuscript for publication.

Thank you very much for your nice comments.